# Do Microplastics and Nanoplastics Pose Risks to Biota in Agricultural Ecosystems?

Naga Raju Maddela [1,2], Balasubramanian Ramakrishnan [3], Tanvi Kadiyala [4], Kadiyala Venkateswarlu [5] and Mallavarapu Megharaj [6,7,*]

1   Departamento de Ciencias Biológicas, Facultad de Ciencias de la Salud, Universidad Técnica de Manabí, Portoviejo 130105, Ecuador
2   Instituto de Investigación, Universidad Técnica de Manabí, Portoviejo 130105, Ecuador
3   Division of Microbiology, ICAR-Indian Agricultural Research Institute, New Delhi 110012, India
4   Lubbock-Cooper High School, 910 Woodrow Rd., Lubbock, TX 79423, USA
5   Formerly Department of Microbiology, Sri Krishnadevaraya University, Anantapuramu 515003, India
6   Global Centre for Environmental Remediation (GCER), College of Engineering, Science and Environment, The University of Newcastle, Callaghan, NSW 2308, Australia
7   Cooperative Research Centre for Contamination Assessment and Remediation of the Environment (CRC CARE), The University of Newcastle, Callaghan, NSW 2308, Australia
*   Correspondence: megh.mallavarapu@newcastle.edu.au

**Abstract:** The presence of microplastics and nanoplastics (MNPs) in soils is becoming pervasive in most agroecosystems. The recent estimates suggest that the soil burden of MNPs in the agroecosystems is more than 0.5 megatons (Mt) annually. In all the agroecosystems, the transformation, migration, and transferring of MNPs, along with other contaminants, and the trophic transfer of MNPs can threaten the food web. MPs can exhibit negative and positive effects, or none, on the physical/chemical properties of soil, soil microbiota, invertebrates, and plant systems, depending on the polymer compositions, additives, and exposure time. Difficulties in comparing the studies on the effects of MNPs, as well as the discrepancies among them, are mostly due to variations in the methods followed for sampling, detection, quantification, and the categorization of particles, abundance, and exposure time. Since agricultural soils are important environmental reservoirs for diverse chemicals and contaminants, they provide milieus for several types of interactions of MNPs with soil biota. The present review critically examines the sources and transformation of MNPs in agricultural soils, the release and fate of additives, as well as their role as vectors of other potential contaminants and influence on soil physical/chemical properties, toxicities to soil biota (i.e., microorganisms, invertebrates, and plants), current regulatory guidelines for the mitigation of MNPs, and future research directions.

**Keywords:** microplastics; nanoplastics; soil biota; toxicity and fate; plant uptake; regulatory guidelines

## 1. Introduction

Significant anthropogenic changes to the Earth's land surface include agricultural activities, which have affected almost 32% of the global land area between 1960 and 2019 [1]. The geographically diverging processes, such as afforestation and cropland abandonment, have been increasing in the Global North, while deforestation and agricultural expansion are on the rise in the Global South. In addition, the distribution of land use between livestock and crops for human consumption is unequal, with only 23% of agricultural land used for crop cultivation. China has the largest agricultural land extent (about 500 mega hectares, Mha), while India has the largest cropland area (about 170 Mha). Though the expansion and intensification of agriculture have increased crop yields significantly, agriculture threatens global biodiversity, with about 24,000 species being threatened with extinction [2]. The global land use dynamics with the critical components of space, time,

and theme, depending on the combination of climatic, edaphic-soil conditions and socio-economic drivers, are essential to tackle sustainability challenges such as food security and environmental safety. The risks for ecological health have increased due to soil pollution by a wide array of contaminants, such as pesticides, mineral fertilizers, sewage sludge, manure, plastic materials used in mulching, greenhouses, tubes for irrigation, packaging, compost with different contaminants, and rural or urban waste, although more so due to plastics in recent times.

Both point-source pollution and diffuse pollution make the agricultural soils a secondary source of pollutants to the surface waters, groundwater, and, subsequently, to oceans and the atmosphere. Tang et al. [3] reported that the pesticide pollution risk was about 64% in global agricultural land by the global model mapping of 92 chemicals commonly used in agriculture in 168 countries. About 34% of the high-risk areas were in high-biodiversity regions. Microplastics (MPs) of different morphology, color, and ecotoxicity that come from various organic polymers and are blended with different additives represent a diverse suite of contaminants [4]. Hence, an assessment of the fate, persistence, and toxicity of this complex, emerging global contaminant suite in agricultural ecosystems must consider their diversity and complexity.

The term "plastic," which originally meant "pliable and easily shaped," has been now used for a chain of synthetic polymers [5], but "plastic" is historically known as "celluloid" and "bakelite." The mass production of synthetic plastics has continually increased since the 1950s. The current global production is about 370 Mt (Megatons; million metric tons), reflecting a similar rise in plastic waste generation. In 2050, the global annual primary plastic production can reach about 1-billion metric tons [6]. Interestingly, China is the world's largest producer of plastics (31% of the overall production), while other Asian countries, including India, synthesize about half the amount of plastics in the world. Plastics are used in different sectors: packaging (~40%), building and construction (~20%), automotive industries (~10%), electrical and electronic industries (~6%), agriculture (3 to 4%), household, leisure, sports (~4%), and others (~17%; this includes plastics for furniture, medical applications, machinery and mechanical engineering, technical parts, etc.) [7]. Although direct usage of plastics in the agricultural sector is lesser than in other sectors, the accumulation of plastics in the agricultural soils is steadily increasing due to intentional uses and unintentionally contaminated water and manure [8]. Almost half of the global usage of plastics in agricultural production is in Asia.

The intensive global production of resins and fibers, with an increase of 8.40% in compound annual growth rate (CAGR) from 1950 (2 Mt) to 2015 (380 Mt) [9], resulted in the release of enormous quantities of plastic wastes from different sectors. Nearly no or slow degradation rates are the key drivers of plastic accumulation in the environment. One of the important receivers of plastics in the environment is agricultural soil. By 2050, global projected plastic waste generation is expected to reach 26,000 Mt, of which 45% is not expected to be recycled or incinerated [9,10]. Based on the size, these particles can be of two types: MPs and nanoplastics (NPs), which have sizes can that be <5 mm and <100 nm, respectively [11]. Plastics of a size >5 mm are considered "macroplastics". In the present review, MPs and NPs together are referred to as MNPs. Individually, the terms MPs and NPs are used to emphasize the size effects. According to a mass balance approach study, farmlands' soil burden of MNPs in the US alone is 70 kilotons $yr^{-1}$ [12]. In intensive farmland soils, the load of different polymers is up to 43,000 particles $kg^{-1}$ [13]. This alarming situation demands great public and political attention to regulate and mitigate plastic waste accumulation [14,15].

The soil systems have been given less importance than the aquatic systems concerning the global efforts to source-track and regulate MNPs (Figure 1a). Nevertheless, research publications related to "MNPs *versus* soil" have increased by >14% from 2018 to 2022 (as of 30 October 2022). The percentage of MNPs generated and accumulated (i.e., 75.60%) in municipal solid waste landfills in the USA (reported in 2018, Figure 1b) and the occurrence of MNPs (either in the form of pieces or particles) in farmland soils in different

countries (Figure 1c) suggest that MNPs pose a great threat to agricultural ecosystems. In the recent past, several investigations focused on MNPs versus farmland (agricultural) soils and revealed insights on MNP distribution and microbial community characteristics [16], the distribution behavior of MPs by FTIR analysis [17], migration characteristics of MPs [18], seasonal variations from MPs with co-contaminants like Cd [19], the effect of MPs on plant growth and soil health [20], and the ecotoxicity of MPs [21]. Other studies focused on the influence of MPs on rhizosphere microbial community structure [22], MPs in organic fertilizers [23], the toxicity of MPs to plant root cells [24], the adsorption and desorption propensities of MPs on to heavy metals like Cd [25], transport pathways of MPs from agricultural soils to the aquatic ecosystems [26], MPs versus arbuscular mycorrhizal fungi [27], the effects of NPs on plant functionalities [28], and many others. These insights indicate the severity of the contamination of MNPs in the farmland soils and highlight the significant burden of MNPs in the agroecosystems that pose considerable threat to the terrestrial food chain.

Recently published reviews are particularly concerned with the extraction and identification methods of MNPs in soils [29], meta-analysis of the literature on the effects of MNPs toward soil invertebrates [30], critical view on MNPs versus soil fauna [31], biosolids as a source of MPs and other pollutants [32], and sources of MPs, their effects, and fate [33]. In addition, other available reviews dealt with the occurrence of MNPs in non-agricultural soils [34–36] and aquatic ecosystems [37–40]. However, information related to different sources of MNPs that enter agricultural soils, their fate, direct and indirect effects toward soil biota, and regulations for mitigations of MNPs is not readily available. Moreover, because of the complexity of the agroecosystem, cultivation methods, different crops cultivated, and agronomic practices, the behavior of MNPs in farmlands could be different compared to that of non-agricultural soils and aquatic ecosystems. Thus, the aim of this review was to provide comprehensive insights on the sources of MNPs in agricultural soils and their transformation, the release, the fate of additives from MNPs, MNPs as vectors of contaminants, the influences of MNPs on soil physical/chemical properties, the toxicity of MNPs to biota (i.e., microbial diversity, invertebrates, and plants), the regulations for the mitigation of MNPs in agricultural soils, and future research directions. Such insights not only provide an understanding about the problems associated with MNPs, but also help in designing effective mitigation strategies, as well as revise or set new regulations for the control of MNPs in the agroecosystems.

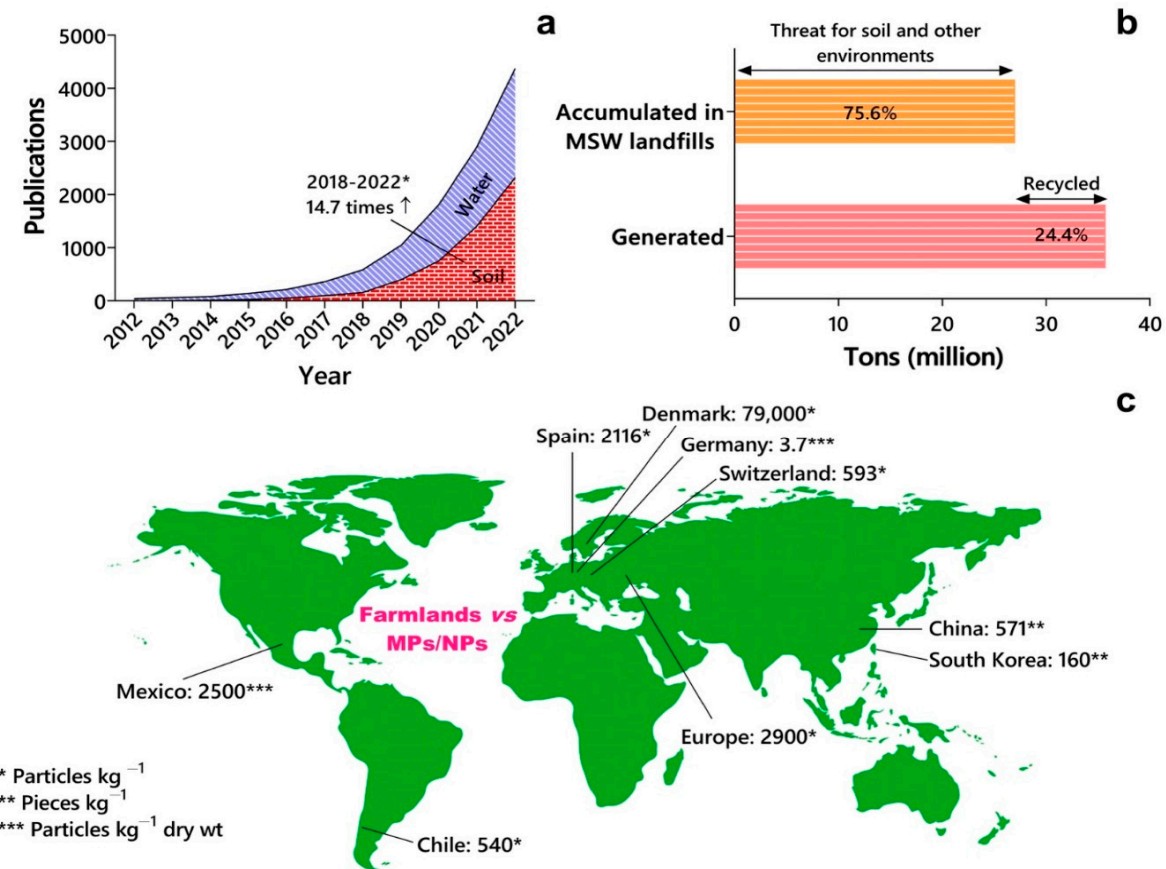

**Figure 1.** Microplastics (MPs)/nanoplastics (NPs)—a global scenario. (**a**) Publication of scientific articles from 2012–2022 *. Data collected from ScienceDirect using "microplastics + water" and "microplastics + soil" keywords (* as of 30 October 2022). (**b**) Amount of MPs generated and accumulated in municipal solid waste (MSW) landfills in USA in 2018 [41]. (**c**) Occurrence of MPs in farmland soils in different countries [36,42].

## 2. Sources of MNPs in Agricultural Soils

Plastics are omnipresent in modern agricultural farms and food production systems. Annually, the agricultural value chains and food packaging use about 12.5 Mt and 37.5 Mt of plastics, respectively [43]. Crop and livestock production accounts for more than 80% of plastic use, followed by fisheries and aquaculture (18%) and forestry (2%) in the agricultural value chains. The plastic protective films (e.g., fumigation, silage, and bale wrap films), the protective films for mulching, nursery, wind tunnel, greenhouse, direct cover and non-woven floating cover, nets (ant-hail, anti-bird, wind breaking, and shading), twine, and pipes for irrigation and drainage are extensively used in the production of agricultural and horticultural crops and livestock. The breakdown process of plastics begins with the action of handling, soil abrasion, water, wind, and UV light. The breakdown products of varying sizes, including macroplastics, MPs, and NPs can last long in these production systems.

The essential sources of MNPs in agricultural ecosystems are the standard practices, such as the application of biosolids (processed sewage sludge) [12] and compost [44], the use of plastic mulching films [45], water pipes, plastic greenhouse covers [44], polymer-based fertilizers [46], and pesticides [47]. Aerial deposition and transport from landfills are other considerable sources of MNPs in agricultural soils [48]. The plastic film mulching technology is popular among farming communities to conserve soil moisture, regulate soil temperature, and prevent weed growth. Light-density polyethylene (LDPE) has been applied to millions of hectares of agricultural soils across the world [49]. Polyvinyl chloride (PVC), as the plastic film, has improved water use efficiency and crop growth yield. However, the mechanized cultivation and the use of thin film generate higher residual

levels of plastic mulch, which disturb the moisture and nutrient transport, decrease seed germination, hinder root growth, induce salinization, and accumulate harmful chemicals such as phthalate esters, aldehydes, and ketones in soils [50]. Thus, the "white revolution" of plastic film mulch technology is becoming "white pollution" in the agricultural production systems. The dumping of municipal wastes in open farm fields, parks, or landfills has been an important factor in spreading MPs to soils in several tropical and subtropical countries [51]. Though many factors like wind and water are involved in the removal of plastics present on the soil surface, substantial MNPs are expected to reach the soil layers [52]. Nearly 40% of MNPs that reach agricultural soils cannot be recovered. Instead, these contaminants break down into a continuum of smaller fragments [53]. The farm soil burden of MPs in Europe and North America were 63,000–430,000 and 44,000–300,000 tons $yr^{-1}$, respectively, and this situation has been attributed to a greater (~50%) use of processed sludge in agriculture [54].

Plastic contaminants can enter the agricultural production systems from damaged, degraded, or discarded agricultural plastic products, and the leakage from non-agricultural sources such as contaminated water, air, and waste. The available technologies remarkably control the entry of MNPs into aquatic systems, but not into soil systems. The current wastewater treatment (WWT) plants can remove ~99% of MNPs from wastewater, which results in mitigating aquatic systems' pollution from MNPs [44]. However, the recovered MNPs remain in the sludge [55], contaminating the agricultural soils upon the sludge amendments [56]. In Portugal, >87% of biosolids generated from WWT plants are applied in the farmlands [57]. A similar practice is also evident in the European Union, where 4–5 Mt sludge solids are applied to the farmlands as fertilizer [51]. About 50% of sewage sludge is processed for agricultural applications in Europe and USA [54]. Thus, wastewater treatment facilities are considered as principal routes of MP entry into the soil systems [58]. The transfer of MPs from urban wastewater to agricultural ecosystems through biosolids has not received enough attention from scientists and regulators [54]. Sludge application (sewage sludge directly) to agricultural soils is common in several countries worldwide [59]. There was a clear correlation between elevated sludge application rates per unit area of farmland and the soil burden of MPs [54]. In some regions, there are environmental laws for farmland soil amendments with biosolids; for instance, only 90 tons of dewatered sludge $ha^{-1}$ $yr^{-1}$ is permissible by Chilean regulations [44]. Concerning compositing, though there is a removal of large- and medium-sized plastics, the problem of generating secondary MNPs through the milling process persists [60]. Nevertheless, the quantities of MP burden and the retention of MPs and transport in soils are impacted by many factors, which include substandard waste management practices, littering, physical properties of MPs (e.g., form, size, and density), the intensity of rainfall and wind speed, and topography [61–63].

In an analysis of MPs present in the sewage sludge and biosolids collected in different countries at different levels of treatment [64], the lowest and highest concentrations of MPs in biosolids reported were found to be 1.0 MPs $g^{-1}$ biosolids [65] and 169,000 MPs $g^{-1}$ dry weight (dwt) sludge [66], respectively. MPs were detected in all types of sludge samples—sewage, primary and secondary, digested, dewatered raw, activated, after-lime-stabilization, and sludge-cake, which means that any sludge or biosolids retain significant amounts of MPs [64]. In addition, common processing steps, such as drying, pasteurization, composting, etc., that are used to produce biosolids for agricultural use do not reduce the MP load in biosolids [54]. Synthetic fibers are the predominant (63% [67], and 92% [44]) type of MNPs present in sludge-amended agricultural soils. Significant levels of synthetic fibers (1.25 [56], and 3.50 picogram (pg) $g^{-1}$ [44]) have been detected in agricultural soils after 1–5 years of sludge amendments, implying that the MNPs remained in the soils for extended periods. Leu et al. [67] found 0.28 pg $g^{-1}$ MPs in farmland soils near Shanghai (China), of which only 28% occurred in the topsoil, which indicates that a significant portion of MNPs seep into deeper layers of the soil.

Polymer coatings for fertilizers for improved nutrient use efficiency and reduced risk of runoff and emissions, and those for seeds for improving germination, serve as the primary

sources of microplastics, in addition to plastic mulch films in modern agriculture [68,69]. The widespread and regular application of polymer-coated fertilizers and seeds in the soil systems will eventually change the physical, chemical, and biological processes. In addition, the fertilizer sacks and pesticide containers are mostly plastic products as they offer increased safety and are easy to transport and store. Plastic trays and food contact films used for consumer packs for distribution and retail to reduce food loss and conserve quality are also discarded in agricultural soils. Though the annual global use of plastic mulch films in crop production is estimated at 7.4 Mt, data on other plastic products used in other phases of agricultural value chains are challenging to estimate [70]. The plastic pollution pathways can vary in different locations, and the identification of the major pathways is critical to find the high-risk areas for further soil sampling and analysis.

Most agricultural plastics are single-use products; with short life spans, they become waste within 12 months. For example, the plastic mulching films decompose due to weathering, and the microplastics that fall off from them remain in the soil. Improper disposal and mismanaged waste plastics will increase soil pollution by secondary microplastics. The abundance of MPs and NPs can vary depending on the sampling sites and other factors related to plastic usage (agricultural intensification) and climate. The abundance values of plastics at the sampling sites can serve only for reference. The types of plastics and additives used for different purposes make toxicological and risk assessments a necessity. In quantitative terms, the main polymers used for the agricultural sectors are polyethylenes (PEs) of both low- and high-density, polypropylene (PP), and PVC, followed by others such as expanded polystyrene (PS), ethylene-vinyl acetate copolymer, and polyethylene terephthalate (PET) [71]. The numerous types of polymers and the additives used in agricultural plastics present a high degree of variability in the toxicological properties and the risks of MNPs to plants and animals. At the end of each crop growing season, the plastic migration begins from soil to aquatic environments through erosion and surface runoff.

## 3. Transformation of MNPs in Farmland Soils

The transformation of MPs, which can be due to mechanical, chemical, and biological processes, is slow, by decades or hundreds of years, and is poorly understood in the agricultural ecosystems [72]. In the heterogeneous and porous medium of soils, the MPs and NPs can aggregate homogenously or heterogeneously with other solid particles. The homo-aggregation of MNPs can be described by the DLVO (Derjaguin, Landau, Verwey, and Overbeek) theory, which states that the interaction force between particles is due to van der Waals and electrostatic forces. The presence of heavy metals and other organic contaminants facilitates the hetero-aggregation of MNPs by sorption. The hetero-aggregation of MNPs with heavy metals and adsorption/desorption with organic pollutants are more than the homo-aggregation in soils with increased environmental risks. The exposure of plastic particles to light (UV rays), high temperature, and oxygen concentrations on the soil surface can accelerate photoaging, which can change the physical/chemical properties, especially the release of additives and monomers. The aggregated and aged MNPs in soils may get absorbed, degraded, and transported by biological organisms, wind, or runoff. Uptake, ingestion, biomagnification, and biodegradation are the major processes of the soil organisms by which MNPs undergo significant changes in the agricultural soils.

The degradation and transformation rates of MNPs in soil environments are minimal, and MNPs highly persist in freshwater, marine, and terrestrial environments [73]. There was a 0.40% degradation of PP after one year of soil incubation, whereas PVC was unaffected even after 10–35 years of soil incubation [74]. Under laboratory-simulated conditions, a maximum of 14–16.70% of PE strips were degraded in 5–5.50 months [75]. Compared to the MPs of PE and PP, PS was transformed easily, as confirmed by an indoor simulated weathering experiment [76]; this is attributed to a faster transformation propensity (likely to crack and break) of PS by friction heat. Thus, the degree of the physical transformation of MPs is greatly influenced by the type of constituents that they have. The physical

transformation of MNPs in the agroecosystems is quite common by mechanical tillage and crop rotation; both accelerate the fragmentation of MNPs in soils [52].

The photo- and thermally initiated oxidative degradations are the primary types of abiotic processes, resulting in the transformation of MNPs in the soil environment. UV radiation and thermal oxidation cause several changes in MNPs in the soil ecosystem [77]. For instance, tensile strength, hydrophobicity, contact angle, and molecular weight were reported to decrease. In contrast, surface roughness, micro-cracks, crystallinity, polarity, functional groups, carbonyl index, leachates, and sorption capacity increased with aging [77]. The soil texture and composition are essential determinants in the degradation of MNPs in the soil environment. Due to higher organic matter content, polymers' degradation in clay soils is more rapid than in sandy soils [78]. On the other hand, chemicals, including those from binder additives, can also induce physical transformation in MNPs.

Sodium sulfide at 0.10 mM caused a reduction in the particle size, cracks, increased roughness, and specific surface area in MPs [10]. Sulfide makes MPs easier to break by causing a chain fracture of the polymer [79]. Furthermore, sulfide treatment also causes an increased O/C ratio on MP surfaces, changes in the distribution of O functional groups, and the appearance of C–O, C=O, and C–S. Overall, sulfide oxidation triggers the formation of the reactive oxygen species (ROS) that causes the oxidation of MNPs [10]. When the MPs of PS were subjected to persulfate oxidation and UV irradiation, there was an increased surface roughness and an increase in oxygen-containing functional groups (e.g., COOH, COOC) [80]. In general, chemical changes in the MNPs during the transformation are introduced mainly due to the breaking, branching, and cross-linking of C–C and C–H bonds in molecules, and the addition of oxygen-containing functional groups [81]. Several free radicals, such as the hydroxyl radical ($\bullet$OH), superhydroxide radical (HO$_2\bullet$), alkoxy radical (RO$\bullet$), and peroxy alkyl radical (RO$_2\bullet$), are significant to the chain initiation, propagation, transfer, and termination during the chemical aging of MNPs [82].

Among the soil biota, microorganisms and soil invertebrates are responsible for the transformation and degradation of MNPs. The selected microbial strains, with the potential for degradation of commonly used plastics, are provided in Table 1. *Bacillus* sp. strain 27 and *Rhodococcus* sp. strain 36 caused 4.0 and 6.40% of weight loss in MPs of PP after 40 days of incubation, respectively; both the strains were native to mangrove sediments [83]. The removal constants (*K*) for PP in *Rhodococcus* sp. strain 36 and *Bacillus* sp. strain 27 were 0.002 and 0.001 day$^{-1}$, respectively [83]. Several microbial species, such as *Paenibacillus* sp. [84], *Pseudomonas*, *Bacillus*, *Brevibacillus*, *Cellulosimicrobium*, *Lysinibacillus*, and *Aspergillus flavus* [85], have been identified as potential degraders of MNPs. The dry weight of PE MPs was reduced by 14.70% after 60 days in non-carbonaceous basal medium inoculated with *Paenibacillus* sp. [84]. Fungal species are efficient in the degradation of MNPs; the reason is that fungal mycelium can adhere or even penetrate the MNPs and promote their degradation by forming chemical bonds (carbonyl, carboxyl, and ester group) and, subsequently, lowering hydrophobicity. Certain microbial enzymes are known to transform MNPs. For instance, oxidase, hydrolase, peroxidase, amidase, and laccase can transform polymers into monomers [86]. The mechanical properties of low-density polyethylene (LDPE) were changed by 27% after 150 days of treatment by *Pleurotus ostreatus* [86]. In addition, different microorganisms use different enzymes to degrade the MNPs. Proteases are the principal enzymes in the degradation of MNPs by *Bacillus* spp. and *Brevibacillus* spp. [87], whereas lignin-degrading fungi use laccases for degrading plastics [88].

**Table 1.** Degradation of commonly used synthetic plastics by selected microorganisms in soils.

| Plastic Type | Microbial Strain | Incubation Time (Days) | Weight Loss (%) | References |
|---|---|---|---|---|
| Low-density polyethylene (LDPE) film | *Rhodococcus ruber* C208<br>*Bacillus* sp. SM1<br>*Ralstonia* sp. SKM2 | 30–180 | 4–180 | [89,90] |
| High-density polyethylene (HDPE) film | *Achromobacter*<br>*Alcaligenes faecalis*<br>(MK517568) | 40–150 | 5.80–9.40 | [91,92] |
| Polystyrene (PS) film | *Xanthomonas* sp.<br>*Rhodococcus ruber* C208<br>*Microbacterium* sp. NA23<br>*Paenibacillus urinalis* NA26<br>*Bacillus* sp. NB6<br>*Pseudomonas aeruginosa* NB26<br>*Rhizopus oryzae* NA1<br>*Aspergillus terreus* NA2<br>*Phanerochaete chrysosporium* NA3<br>*Exiguobacterium* sp. RIT594 | 8–56 | 40–56 | [93–96] |
| Polypropylene (PP) film | *Pseudomonas stutzeri*<br>*Bacillus subtilis*<br>*Bacillus flexus*<br>*Phanerochaete chrysosporium*<br>*Engyodontium album*<br>*Lysinibacillus* sp. JJY0216 | 26–356 | 0–5% | [97–99] |
| Polyvinyl chloride (PVC) plasticized film/sheet | *Penicillium janthinellum*<br>*Mycobacterium* sp. NK0301<br>*Pseudomonas citronellolis*<br>*Trichoderma hamatum*<br>*Bacillus amyloliquefaciens* | 45–300 | 0–33% | [100–103] |
| Polyurethane (PU) foam and film | *Corynebacterium* sp. B12<br>*Pseudomonas aeruginosa*<br>*Comamonas acidovorans*<br>*Alternaria* sp.<br>*Penicillium* sp.<br>*Aspergillus* sp. | 7–84 | 1.20–17.70 | [104–106] |

## 4. Release and Fate of Additives from MNPs

Various chemical substances are added intentionally during the production and processing of plastics. For example, the additions of antioxidants, plasticizers, and flame retardants improve and impart specific properties, while polymerization catalysts, solvents, or lubricants are used as processing aids. In addition, byproducts, breakdown products, and contaminants are the unintentionally added substances in plastic products. More than 10,000 chemical substances have been identified with varying persistence, accumulation, and toxicity levels from scientific, industrial, and regulatory data sources [107]. Thus, the MNPs contain several chemical congeners/additives, such as dioxins, polycyclic aromatic hydrocarbons, heavy metals (e.g., lead, tin, and cadmium), phthalates, brominated flame retardants, bisphenol A (BPA), and BPA dimethacrylate [108]. Such additives are often mixed to expand final plastic products' utility and specific properties. Several of these additives (e.g., BPA and nonylphenol) are endocrine-disrupting chemicals (EDCs) [109]. The most detected EDCs in the leaching from particulate plastics were estrogen, BPA, bisphenol S, octylphenol, and nonylphenol; the second-most-often detected EDC was plastic additive, i.e., BPA with an identified mean concentration of $475 \pm 882$ µg kg$^{-1}$ [110]. The current

research efforts are focused more on the well-known hazardous chemical substances but less on MNPs.

The additives in plastics can migrate from the plastics to the surrounding medium (i.e., air, water, or soil) they are in contact with, or they can also migrate within the plastics [111]. Additives are covalently or non-covalently linked with plastics, and these chemicals are easily leached from the source materials subjected to the environmental deterioration [112]. Desorption and UV-degradation are critical mechanisms in the leaching of additives from MNPs. Nevertheless, the additives are released from the parental material at any time, i.e., the production phase, use phase, and end-of-life phase [108]. The plastic waste that is subjected to landfilling and littering is subsequently degraded (i.e., chemical, mechanical, and biodegradation) and then fragmented to MPs or mineralized to $CO_2$ or inorganic molecules [108]; in all these events, there is a possibility for the release of additives from plastics. Notably, the composition of additives decides which chemical is to be released first during leaching. However, the migration potential is also essential in the leaching of additives, i.e., availability versus solubility behavior. Generally, polymers have three-dimensional porous structures inside, allowing additives to migrate. Therefore, more minor additives move freely in the polymers containing larger porous structures [113]. In addition, the hydrophilicity and hydrophobicity of additives influence their leaching from their source material. For example, dimethyl phthalate (DMP) is more easily released from resin-based polymers than diethylhexyl phthalate (DEHP) because DMP is hydrophilic, whereas DEHP is hydrophobic [114]. The order of appearance of different phthalic acid esters released from the phthalate-containing product in landfill leachate was as follows: hydrophilic phthalic acid diester (PADE) > hydrophilic phthalic acid monoester (PAME) > moderate hydrophobic PADE > moderate hydrophobic PAME > hydrophobic PADE > hydrophobic PAME. Still, the concentrations of all three PAMEs were higher than the concentrations of the respective PADEs [115]. These results suggest that PADEs are more readily released from their source materials than PAMEs.

The type of degradation products released depends on multiple factors, such as polymer type, the degradation mechanism, and environmental factors (i.e., temperature and oxygen) [116]. Even by a single degradation mechanism, different polymers release different additives. For example, during thermal degradation, nitrogen-containing polymers (e.g., nylons, polyacrylonitrile, PU) release hydrogen cyanide (HCN), but chlorine-containing polymers (e.g., PVC) release hydrogen chloride and dioxins [117]. Since the chemical substances that are released from the plastics may have different hazardous properties, such as endocrine disruption, carcinogenicity, mutagenicity, chronic, acute toxicity, bioaccumulation potential, and persistence, their fate depends on several soil factors and the exposure potential to the biota for their uptake, ingestion, and biomagnification.

## 5. MNPs as Vectors of Other Contaminants

The contaminants, such as hydrophobic- and persistent organic compounds, heavy metals, and microbial pathogens, are adsorbed onto MNPs as hetero-aggregates in the soils. Hence, the MNPs act as potential vectors by bringing different contaminants (e.g., pharmaceuticals, polyaromatic hydrocarbons, agrochemicals, and engineered nanomaterials) to agricultural soils [77,118]. Thus, there is a significant concern about the co-transport of MNPs with the sorbed pollutants of farm soils. MNPs relocate the immobile contaminants, which have a strong propensity to interact with the soil matrices [118]. Thus, the hetero-aggregates of MNPs threaten the groundwater resources [119]. Several characteristics of MNPs, such as type, specific surface area, porosity, number of adsorption sites on the surface, and hydrophobicity, as well as soil physical/chemical properties like pH, ionic strength or salinity, texture, and metal cation concentration, influence the initial adsorption of contaminants by MNPs [77]. Regarding the transportation of organic contaminants along with MNPs, the critical conditions include a high abundance of particles, the type of contaminants, a greater mobility of particles than the contaminants, and a lower desorption of contaminant from the MNPs during the travel time of MNPs [118]. However, even if the

first three conditions are met, the desorption rate of the contaminant is a deciding factor in the MNP-mediated contaminant transport. Furthermore, how long a particle carries the contaminant in the soil matrix varies between MPs and NPs. In general, particles are mobile at a diameter of one μm [120].

In the soil matrix, transporting more prominent MPs and fibers requires larger soil pore diameters or preferred flow paths. On the other hand, NP transportation is limited by a diffusion to the surface of the collector [121]. The MNP-facilitated contaminant transport can be explained in terms of the Damköhler number (DN), which is the dimensionless ratio of a characteristic liquid residence time to the reaction time) [122]. If the DN value is <0.01, it indicates a fully decoupled transport, where the contaminant desorption rate during MNPs' travel time is negligible, and there is a feasibility for the MNPs-mediated transport of the contaminant. In contrast, if the DN value is >100, it indicates a full equilibrium condition, where MNPs do not contribute to the contaminant transport. Upon the exposure of polyethylene MPs to air with UV light, the carbonyl index (CI) increased from 0.07 to 0.62 (~9 times hike) [123]. Generally, the UV exposure of MPs results in forming free oxygen radicals that can induce the formation of oxidative functional groups [124]. Thus, the propensity of MNPs to serve as vectors is enhanced by forming such functional groups. The aging of MPs causes an increase in surface area and decrease in particle size, and it introduces several O-containing functional groups (e.g., C, =O, –COOH, and –OH groups), which enable MPs to attract several environmental pollutants and act as vectors [125].

MNPs play a significant role as vectors in the accumulation and transport of heavy metals (HMs), organic contaminants, and engineered nanomaterials. HMs are adsorbed to the MNPs' surface by chemical interactions, such as electrostatic attraction, surface complexation, and precipitation. There was an adsorption of 0.091 ($Cu^{2+}$) to 0.470 ($Pb^{2+}$) $mg\ kg^{-1}$ of HMs on PP and PE-type MPs in the aquatic environment through electrostatic attraction [126]. Guan et al. [127] found 819.90 $mg\ kg^{-1}$ $Cu^{2+}$ on PS microplastic through electrostatic adsorption. The adsorption rates of $Pb^{2+}$ with different MPs followed the order: polymethylmethacrylate > PE > PP. The highest adsorption of $Pb^{2+}$ was 4.21 $mg\ g^{-1}$ by polymethylmethacrylate [128]. The in vitro studies with aquatic matrices revealed that salinity and pH play a significant role in the adsorption of HMs by MNPs. A higher salinity reduced the HMs adsorption by MPs [129], a higher pH induced a more significant adsorption of Cd onto HDPE [130], and an acidic pH favored the adsorption of Cr(VI) [131]. The variation in the adsorption rates of HMs by MNPs is attributed to the metals' reactivity, ion-exchange capacity, and partition coefficient [132]. Like HMs, several organic pollutants are adsorbed onto the MNPs surface, resulting in the subsequent co-transfer of contaminants to the soils. The organic pollutants that have been co-transferred by MNPs include polycyclic aromatic hydrocarbons (PAHs) [133], pharmaceuticals [134], 17β-estradiol [135], and triclosan [125]. To understand the distribution and recovery of pesticides in the presence of MPs, a soil column experiment was conducted by Ramos et al. [136]. They found that the adsorption of trifluralin, procymidone, and chlorpyrifos to PE was 98.90, 95.90, and 98.60%, respectively (Figure 2), implying the propensity of PE to act as vectors to carry different organic contaminants in the soil ecosystem. Once these organics are bound to MNPs, their recovery is difficult to achieve. For instance, the recovery of other pesticides/herbicides bound to PE was in the range of 0.30–1.20% (Figure 2).

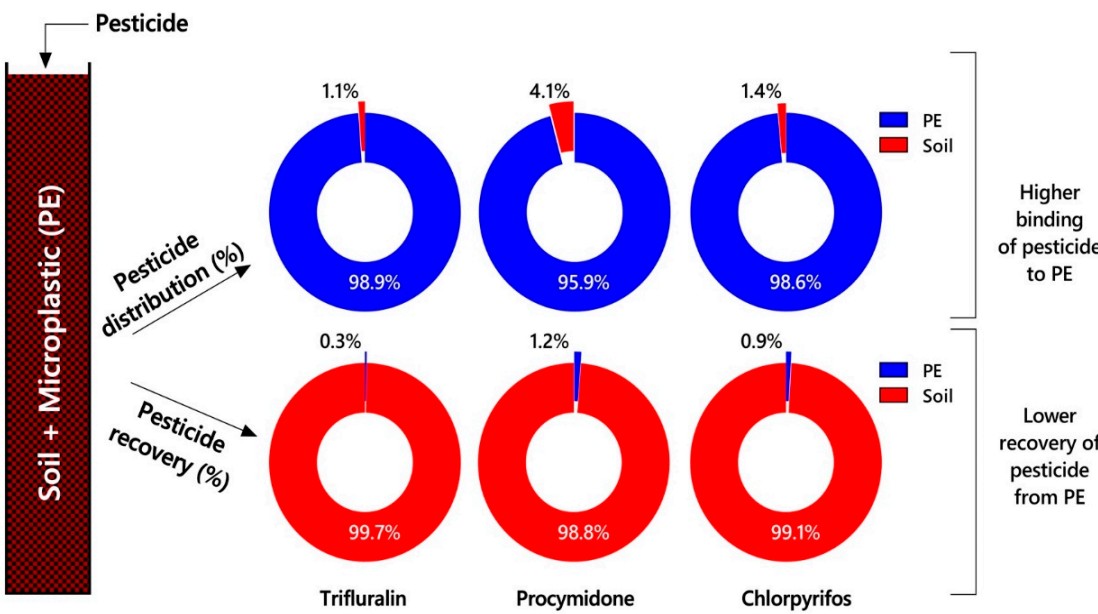

**Figure 2.** Distribution and recovery of pesticides in a soil + microplastics column (based on data from Ramos et al. [136]) (PE—Polyethylene).

The principal mechanisms behind the interactions between organic pollutants and MNPs are the π–π interactions, van der Waals, electrostatic interaction, hydrogen bonding, hydrophobic interactions, pore filling, and cation ligand interactions [77]; structures and properties of MNPs generally determine these mechanisms. On the other hand, the engineered nanoparticles (ENPs) that are released to the soil ecosystem from different sources (e.g., cosmetics, textile industries, and wastewater treatment facilities) are readily adhered to MNPs by physical adsorption (e.g., π–π interactions). A study on the aquatic matrix revealed that an increased aromaticity in MNPs induces the π–π interactions towards ENPs. For instance, PS could adsorb >90% of silver nanoparticles (AgNPs) at a lower AgNPs/PS mass ratio because of the induction of increased aromaticity of the benzene ring on PS [137]. There was an increased transport of $TiO_2$ nanoparticles in quartz sand by MNPs at pH 7.0 [138]. In addition to the chemical contaminants, microorganisms can colonize the MNPs by forming a biofilm. Biofilms thus formed with MNPs can sorb more pollutants than MNPs without the biofilm [123]. However, information on MNPs as vectors for the accumulation and transport of emerging contaminants in the soil environments, particularly in agricultural soils, is still limited. More research is needed to focus on the occurrence and concentrations of MNPs with chemical and biological contaminants, especially plant and human pathogens.

## 6. Influence of MNPs on Soil Physical/Chemical Properties

MNPs seriously affect the soil's physical/chemical properties (e.g., water retention capacity, pore size, pore availability, hydraulic properties, and soil conductivity) (Figure 3a–d) and biological properties, especially those of soil microbial community. At environmentally relevant nominal concentrations (i.e., 2%), polyester, polyacrylic, and polyethylene MPs decreased the soil bulk density and caused changes in the structure and function in a loamy sand soil within 5 weeks [139]. The changes in the soil bulk density are mainly due to the low density of MPs, relative to those of many natural soil minerals. The PS-based MNPs changed the chemical properties and functional groups in dissolved organic matter aromatic rings [140]. Up to 6% of PP MPs (20, 200, and 500 μm) decreased the Ks values (saturated hydraulic conductivity) of loam, clay, and sandy soils by 70, 77, and 96%, respectively [141]. Especially in clay soils, the addition of MPs caused a more significant reduction in water retention capacity than in the loamy and sandy soils [141]. Furthermore, MPs

caused a considerable increase in the slope of the water characteristic curve (SWCC) and a decrease in the saturated water content ($\theta s$) and residual water content ($\theta r$) in clay soils.

Compared to larger-sized MPs, relatively smaller-sized MPs significantly reduced soil porosity and aeration [142]. There is an inverse relationship between the number of MPs and the number of micropores in soil. Zhang et al. [143] found a reduction in pore-size distribution upon mixing MPs with the soil; this subsequently reduced the hydraulic conductivity of saturated soils. MPs can affect the distribution of soil water–stable aggregates and impair water infiltration by decreasing soil stability [144]. Impaired soil permeability and stability adversely affect the vertical growth of plant roots and, subsequently, plant yield. Available results related to the correlation analysis of irrigation impacted MPs migration in a soil column (Figure 3a), the heavy metal content of soil and MPs (Figure 3b), the response of soil properties to MPs (Figure 3c), and soil aggregate size fractions influenced by MPs (polyester microfibers) and soil organic matter suggest that MNPs (Figure 3d) have significant impact on soil physical/chemical properties.

Several recent investigations have established the adverse effects of MNPs toward soil physical/chemical properties, such as structure [139,143,145–148], porosity [143,148,149], bulk density [139,143,150], water content [139,143,150–152], pH [153,154], organic matter [153,155], and nutrients [156,157]. Most of these investigations have been carried out with different types of MPs (polyacrylic acid, polyamide, polyethersulfone (PES), PE, LDPE, HDPE, PET, PP, PS, polyurethane (PU), and polyacrylonitrile) and soils (e.g., loamy sand, sandy loam, sandy silt, clayey loam, clay nitisol, clay, and sandy) at different experimental conditions (MPs dose, 0.05–2% $w/w$, experimental period, 42 days $\text{yr}^{-1}$). However, the available data firmly suggest that the MPs are underlying drivers of changing soil fertility through numerous factors such as decreased water stable aggregates and bulk density, increased soil moisture evaporation and water-holding capacity, interrupted vertical water flow, changed water content of the adjacent soil layers, increased or decreased soil pH and nutrients, and decreased dissolved organic carbon (DOC) and water-soluble organic carbon (WSOC) [158]. In contrast, no significant impacts of MPs on soil physical/chemical properties were also reported in several other investigations [139,143,150,154,159]. Such mixed results on the effects of MPs toward soil physical/chemical properties can be attributed to the parameters of MPs (e.g., polymer type, dose, size, and shape) and soil properties tested.

The properties of MPs are altered upon their entry into the soil matrix. There is a hetero-aggregation of MPs with the soil inorganic elements, such as Fe, Mg, Si, and Al [160]. Having a positive charge, Fe oxides readily interact with the negatively charged MPs by electrostatic attraction. The hetero-aggregation of MPs with soil minerals and OM enhances the density and zeta potential of MPs. Depending on the shapes (fibers, films, foams, and fragments) and polymer types, MPs can change soil pH, with those of foam and fragment shapes decreasing initially and then increasing the pH as observed in the laboratory incubation studies [161]. In another study, Qi et al. [162] reported that the LDPE films increased the pH of soils. There are contradictory reports on the effect of HDPE films on soil pH [144,163]. Hence, the MNPs can have different effects on various soil properties, depending on the size, shape, exposure time, polymer type, and even the "rhizosphere effect" due to the presence of plants. More importantly, the effect of MNPs on the soil properties can change from non-significant to significant levels with their abundance and the exposure time increasing in most agricultural ecosystems.

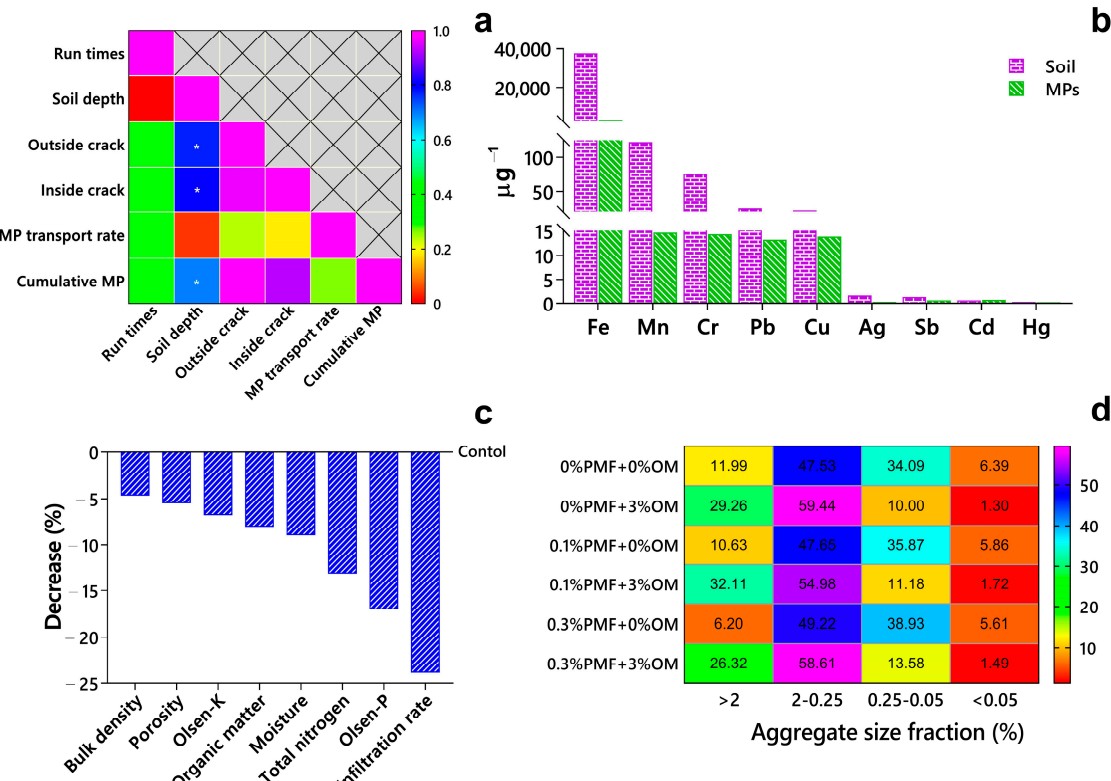

**Figure 3.** Impact of MPs/NPs on soil properties. (**a**) Correlation analysis (rainbow heatmap) of irrigation impacted MPs migration in a soil column (based on data from Zhao et al. [164]) (* indicates negative correlation). (**b**) Heavy metal content (mean values) of soil and MPs (based on data from Zhou et al. [165]). (**c**) Response of soil properties to MPs at concentrations ranging from 0.01 to 640,000 mg kg$^{-1}$ (meta-analysis data from Gao et al. [20]). (**d**) Soil aggregate size fractions (%) as influenced by MPs (polyester microfibers) and soil organic matter (based on data from Zhang et al. [143]).

## 7. Toxicity of MNPs to Different Soil Biota

### 7.1. Microbial Diversity

The ecological interactions between the microorganisms and MNPs are complex and have received enough attention from researchers to identify suitable bioremediating agents. The microorganisms can adhere, colonize, and form biofilms, depending on the surface rugosity and hydrophobicity of MNPs [166]. The MNPs influence the soil microbial community structure, metabolism, and functions. After 90 days of incubation in Cinnamon soil, LDPE (76 mg kg$^{-1}$) caused an alteration in the bacterial community composition [167]. In another study, LDPE, and PVC at concentrations of 1–5% reduced both the richness and diversity of the bacterial community after 50 days of incubation. However, relative bacterial abundance was affected in a community-dependent fashion (i.e., based on phyla and families; increased *Burkholderiaceae* and decreased *Sphingomonadaceae* and *Xanthobacteraceae*) [168]. There was an enrichment of *Acidobacteria* and *Bacteroidetes* in loamy and sandy soils dosed with 2% of PE and PP and incubated for 29 days. However, the same experimental conditions depleted the populations of *Deinococcus-Thermus* and *Chloroflexi* [169]. The richness (Chao1 and ACE) (Figure 4a) and diversity (Simpson's and Shannon's) (Figure 4b) of bacterial communities were significantly affected by MPs in an acid, cropped soil. In an Ustic Cambosol type soil co-amended with PE (1%) and ciprofloxacin, the bacterial community diversity was reduced within 35 days of incubation [159]. However, the members of *Serratia* and *Achromobacter* were abundant in this co-amended soil. The abundance of ammonia-oxidizing bacteria and the copies of nitrite reductase (*nirS*) gene were reduced in clay loam soil treated with LDPE at different concentrations (0.10, 0.50, 1, 3, 6 and

18%) for 30 days [170]. However, MPs also have certain positive effects on soil microbial community structure.

At a concentration range of 0.01–1%, PU showed no significant effects on bacterial diversity. However, PU increased the abundance of *Firmicutes*, *Bacteroidetes*, and *Verrucomicrobia* [146]. Similarly, polyethersulfone (PES) at 0.40% increased the arbuscular mycorrhizal colonization in sandy loam soils after 50 days of treatment [171]. Interestingly, NPs of PS (0.50%) preserved the bacterial diversity and composition by mitigating the adverse effects of sulfamethazine (SMZ) [172]. These findings suggest that MNPs have both negative and positive impact on the abundance and diversity of soil microorganisms. However, most of these investigations have been carried out at the laboratory level with different soils dosed with the pure form of MNPs at environmentally relevant concentrations with a limited incubation period (for weeks and maximum <1 year). Field-level and long-term experiments are greatly warranted to understand the exact effects of MNPs on soil microbial activities. In addition, instead of focusing on the pure form of MNPs, MNP-laden agricultural inputs need to be investigated for their impact on the nonbiologically mediated and biological processes.

In the soil environments, microbial communities, rather than individual members, may apply diverse biofilm-mediated degradation processes involving penetration, leaching, and enzymatic action to transform and degrade MPs/NPs. The combination of biological and nonbiologically mediated processes influenced by the soil properties, such as organic matter and mineral contents, pH, and ionic strength, and the environmental factors, including temperature, rainfall, and irradiation, determine the biotic transformations of MNPs. The abiotic and biotic transformation mechanisms will determine the dispersion, accumulation, and the fate, thus, the lifespans, of MNPs in agricultural soils. By mining the microbial genomes that are publicly available, Gambarini et al. [173] identified the plastic-degrading traits of about 16,170 putative orthologs in species belonging to different bacterial phyla such as *Proteobacteria*, *Actinobacteria*, *Firmicutes*, *Bacteroidetes,* and *Cyanobacteria*, and the fungal members of phyla, such as Ascomycota, Basidiomycota, and Mucoromycota. When metal ion-blended PP were added to the cultures of *Phanerochaete chrysosporium* NCIM 1170 and *Engyodontium album* MTP091 for one year, the gravimetric weight loss and thermos gravimetric analysis (TGA) weight loss were in the range of 9.42–18.80 and 57–79%, respectively [98]. *Alternaria* sp., *Penicillium* section Lanata-Divaricata and *Aspergillus* section *flavi* were found to utilize two polyester polyurethane as a sole carbon source [106]. Aboveground biomass and the colonization of AM (arbuscular mycorrhizal) fungi were increased under polyester microfiber addition [171]. Zrimec et al. [174] used information from the experimentally verified enzymes and metagenomes (soils and oceans) to construct hidden Markov models (HMMs), identify nonredundant enzyme homologues (about 30,000), observe the enrichment of plastic-degrading enzymes within microbial members of α- and γ-*Proteobacteria*, and show the correlation between the country-specific pollution trends and the plastic-degrading potential of the Earth's microbiome. Probably, this is the first evidence of a measurable effect and the adaptation of the Earth's microbiome to the global plastic pollution trends. Information on the novel plastic-degrading enzymes and the diversity of microorganisms with plastic-degrading potential is pertinent to develop new approaches to managing plastic waste.

### 7.2. Invertebrates

Earthworms, which play significant roles in the soil food web, are the central focus of research on the effect of plastic pollution on soil animals. Species of earthworms, such as *Lumbricus terrestris*, *Eisenia fetida*, and *Eisenia Andrei,* can ingest and digest MNPs, with variable effects on the growth rate, reproductive rate, and mortality [175]. The activities of earthworms increase the transport and incorporation of MNPs into the soil matrix. The meta-analysis of the effects of MNPs on earthworms and nematodes showed dose-dependent responses [30]. MNPs, even at 0.10% ($w/w$), can reduce the growth and survival of earthworms. Increased mortality and decreased growth rates in *L. terrestris*

were observed in litter containing higher concentrations (28, 45, and 60%) of MPs of PE than at lower concentrations (7%) and control (0%) [176]. Many functions in soil animals were known to be influenced (decreased/increased) by MPs (Figure 4c). Kim et al. [177] observed more adverse effects of NPs of PS on invertebrates in soil matrix than in liquid media at similar concentrations of PS (i.e., either 10 mg kg$^{-1}$ soil or 10 mg L$^{-1}$ water). In the same investigation, *Caenorhabditis elegans* showed greater sensitivity towards large-sized NPs of PS (530 nm) than smaller NPs (42 nm) [177]. In contrast, smaller PS particles (50 nm) showed more significant toxicity than larger PS particles (0.50 to 6 μm) on the fecundity and life span of rotifer species [178]. Similarly, Ziajahromi et al. [179] observed that smaller PE particles (10–27 μm) more adversely affected the growth and survival of *Chironomus tepperi* than larger PE particles (100–126 μm). The same trend was observed in *C. elegans*, where smaller PS particles (0.10–1.0 μm) showed more toxicity than the larger ones (5.0 μm). The size-based differences in the toxicities of MNPs can be explained by the fact that smaller particles are quickly accumulated, ingested, and digested, relative to lager particles, by the organisms.

The gut microbiomes of invertebrates can be significant sources of microorganisms capable of plastic degradation. The larvae of mealworms (*Tenebrio molitor*) and wax moth (*Galleria mellonella*) cut plastic pieces by grinding, and their gut microbiome with *Bacillus* sp. strain YP1 and the members belonging to *Citrobacter* and *Kosakonia* aided in the degradation of PS and PE [180–182]. MNPs have also been identified as critical factors that adversely affect different functions in soil invertebrates, such as reproductive fitness and success [177], ingestion behavior [183], oxidative stress response [184], locomotion [185], and gene expression [186]. However, the contrasting results on ecotoxicities of MNPs against soil invertebrates were reported, which could be attributed to differences in the properties of both MNPs (i.e., size, shape, and polymer chemistry) and soil invertebrates (i.e., species tested, dose range, and the duration and conditions of exposure) used in different investigations. For example, the particle-to-mouth size ratio, which is species-dependent, is a critical determinant of particle ingestion by soil invertebrates. Hence, diverse species of soil invertebrates will respond contrarily to the MNPs. In a recent study, Yang et al. [187] showed that the combined application of NPs of PS (100 nm-sized particles added at 1000 mg kg$^{-1}$ soil) and tetracycline (20 mg kg$^{-1}$) led to the enhanced toxicity of tetracycline and the enrichment of antibiotic resistance genes (ARGs of vancomycin, tetracycline, macrolide-lincosamide-streptogramin Group B (MLSB) and chloramphenicol) in *Enchytraeus crypticus*. This finding suggests that the ecotoxicity of many co-existing contaminants can threaten the soil microbiome and invertebrates, and thus, the health of agroecosystems.

### 7.3. Plants

Higher plants absorb materials or particles of 3–4 nm in size, or up to 40–50 nm in some cases. Bandmann et al. [188] provided the first report on the uptake of PS nanoparticles of 20–40 nm in size by tobacco BY-2 cells attributed to endocytosis. MNPs in the terrestrial ecosystem are known to be accessible to plant systems and cause phytotoxicities upon root uptake and translocation. Subsequently, MNPs enter the food chain, and, thus, "food safety" becomes an issue. In the recent past, several investigations have confirmed that MNPs are subjected to root uptake and translocation to the edible tissues of several food crop species such as wheat [189], carrot [190], cucumber [191], rice [192], maize [193], and lettuce [194]. In a study using MNPs of PS, the accumulation of MNPs was observed at the root cap cells of *Arabidopsis* and wheat but not uptake into roots [195]. Depending on the size and charge of MNPs and the nature of plant tissues, such as the sticky or hydrophobic surface layer, plants can adsorb or internalize them. Li et al. [196] developed the quantitative, pyrolysis gas chromatography-mass spectrometry (Py-GC-/MS) analysis of NPs involving digestion (alkaline), precipitation (cellulose), and leaching (ultrasonic) with the detection levels of 2.31–4.15 μg g$^{-1}$ and 3.87–8.20 μg g$^{-1}$ for PS and polymethyl methacrylate, PMMA), respectively. When spiked with spherical NPs, cucumber plants grown hydroponically had

0–6893 µg PS g$^{-1}$, after exposure to 50 mg L$^{-1}$ of 100 nm PS for 7 and 14 days. Both the Py-GC/MS and scanning electron microscopy analysis showed the presence of PS in various dried cucumber, suggesting the uptake, translocation, and accumulation of NPs [196]. Nevertheless, the technical barriers in quantifying diverse plastic particles, such as those of PE and PVC, and those other than pristine and spherical particles in the field-grown plants are yet to be addressed. Undeniably, crop plants feed the world, and the MNPs' burden on crop plants causes significant food insecurity. In a meta-analysis study, it has been confirmed that MPs can affect photosynthesis and antioxidant system and morphology in different plants, such as maize, lettuce, wheat, and cress (Figure 4d). The crack-entry pathway is the principal route of MNP uptake, evidently from the hydroponic experiments using wheat and lettuce plants, where PS and PMMA particles penetrated the stele of both species [194]. The confocal microscopic observations confirmed the accumulation of PS at the root surface and entry into the epidermis, cortex, and stele, especially in the xylem of root seedlings of rice [192]. Furthermore, MPs tend to twist and deform the cell walls and can form larger pores, enough to penetrate larger particles [190]. A higher transpiration rate is also a significant factor in enhancing the root uptake of MNPs [194]. On the other hand, apoplastic [192,197] and symplastic [198] transports seem to be the central pathways in the translocation of MNPs to the areal tissues. Nonetheless, no studies have focused on the genetic level to identify the molecular basis of MNPs transport in plants [199].

Several adverse effects of MNPs on plant functionalities have been reported. For instance, a reduction in the root length, the fresh weight of the plant and chlorophyll content [197], decrement in shoot/root ratio [189], induction of genetic changes [200], reduction in seed setting and root/shoot ratio [201], altered metabolic pathways [202], decrement in seed germination rate [203], dry biomass and plant height [204], reduction in the photosynthetic metabolism of leaves, and interference in the mineral nutrition metabolism in the roots, stems, and leaves [205]. Importantly, MNPs can also exhibit indirect negative effects on the plant growth and performance by altering the soil's physical/chemical properties [177], soil microbiome [158], and invertebrates [31]. The charge of the MNPs is also an essential factor in inducing different phytotoxicities. The positively charged PS NPs (PS–NH$_2$) were less accumulated in the root tips in *Arabidopsis thaliana* but caused a higher accumulation of ROS and, thus, affected plant growth and seed development worse than the negatively charged NPs of PS (sulfonic-modified; PS-SO$_3$H) [197]. Conversely, the negatively charged NPs of PS showed a higher accumulation in the apoplast and xylem. Very recently, it was observed that NPs of PS could significantly alter the gene expression pattern in a tissue-specific manner in *Triticum aestivum* L. [200]. In a hydroponic condition, 0.01–10 mg L$^{-1}$ of NPs of PS (100 nm) altered several plant functions, such as carbon metabolism, amino acid biosynthesis, mitogen-activated protein kinase (MAPK)-signaling pathway, plant hormone signal transduction, and plant–pathogen interaction pathways. Cress seed germination was reduced dose-dependently in treatment with MNPs at 10$^{-3}$ to 10$^{-7}$ particles mL$^{-1}$ [203]. When MNPs are with co-pollutants, the phytotoxicities are even worse. For example, photosynthesis and antioxidant activities in rice were adversely affected by the combination of NPs of PS (0.20 g L$^{-1}$) and As(III) (4.0 mg L$^{-1}$) than As(III) treatment alone [206]. However, there are also positive and non-significant effects of MNPs on plant functionalities [199], which makes it difficult to conclude the impact of MNPs on plants.

The MNPs exert variable effects on the biometrical, biochemical, and physiological properties, including cytotoxic and genotoxic effects, depending on the physical/chemical properties of particles, plant species, and the exposure time [199]. The discrepant results on the impact of MNPs toward plants demand future studies that should focus on elucidating mechanisms underlying bioaccumulation, phytotoxicity, and trophic transfer. Trophic transfers of MNPs from plants to humans may be difficult to demonstrate. Still, humans, as one of the major receivers of MNPs, are beginning to be understood as reports suggest the presence of phthalates (plastic additives) in the urine samples of pregnant women [207], microplastics in the feces of healthy human volunteers [208] and infants [209], and mi-

croparticles of PE in the blood samples of healthy adults [210]. The above insights clearly imply that MNPs in farmland soils are subjected to root uptake and translocation into food crops, posing a significant threat to the terrestrial food chain by MNPs in farmlands. However, technical barriers in the quantification of MNPs limit the mitigation of MNPs' entry into farmland soils and their further dissipation in the agroecosystems.

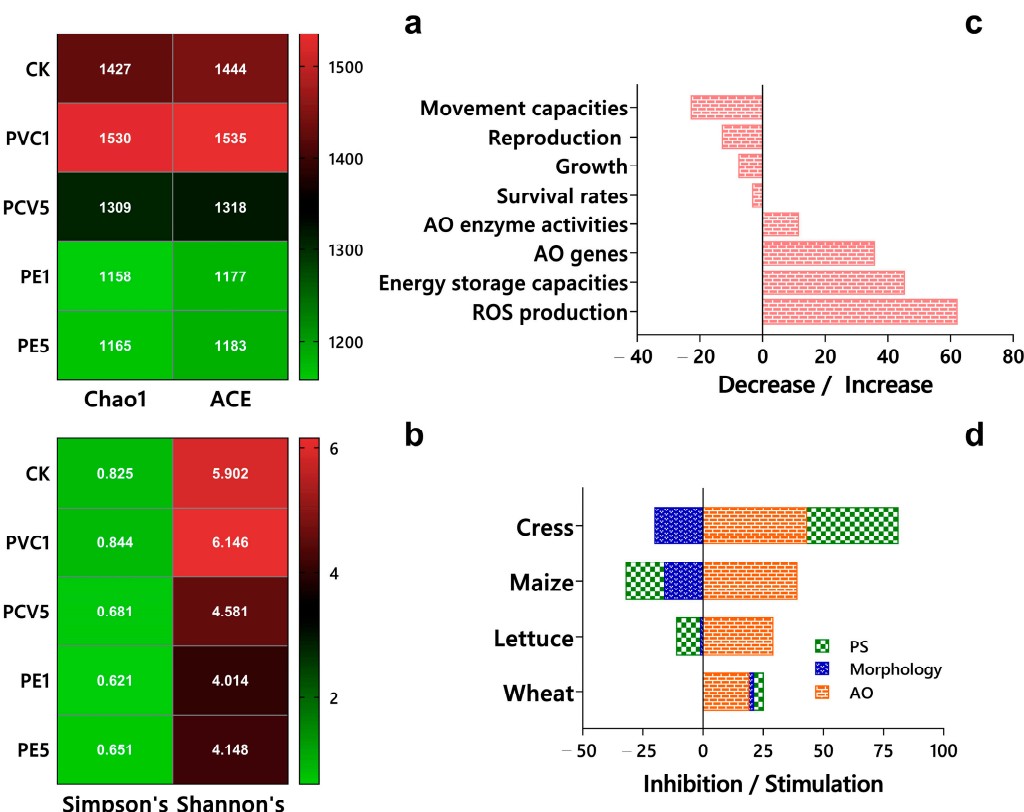

**Figure 4.** Response of microorganisms, plants, and animals to MPs/NPs in farmland soils. (**a**) Community richness (Chao1 and ACE), and (**b**) Community diversity (Simpson's and Shannon's) of bacteria as affected by MPs in soil (based on data from Fei et al. [168]). CK—control soil, PVC1—polyvinyl chloride 1% (*w/w*), PVC5—polyvinyl chloride 5% (*w/w*), PE1—polyethylene 1% (*w/w*), PE5—polyethylene 5% (*w/w*). (**c**) Response of animal functions to MPs in soil, as indicated by meta-analysis [211] (AO—antioxidant, ROS—Reactive oxygen species). (**d**) Impact of MPs on specific functions in different plant species. Per cent values of change were transformed from RR (risk ratio) of meta-analysis data [212], (PS—Photosynthesis, AO—Antioxidant system).

## 8. Regulatory Guidelines for Mitigation of MNPs

Framing regulatory guidelines for controlling MNPs in different environments is a challenging issue. The two main cumbersome reasons are: the intentional/unintentional release of MNPs, and the classification and quantification of MNPs. One of the reasons is that certain MPs are not included intentionally to the products. Instead, the degradation of plastics generates them. In this case, it is challenging to set regulations on unintentionally released MPs, and it would not be possible to regulate such MPs under the existing chemicals legislations. On the other hand, certain MPs are deliberately manufactured and intentionally added during the manufacturing of certain products (e.g., fertilizer coatings, phytosanitary products, cosmetics, household and industrial detergents, cleaning products, cosmetics, and paints, and products used in the oil and gas industries). The European Chemical Agency (ECHA) has confirmed that there is a considerable risk to the ecosystems by releasing such intentionally added MPs if not adequately controlled. In January 2019, the ECHA set regulations on the MPs in the products placed on the EU/EEA markets to prevent the release of 500,000 tons of MPs over 20 years (annual use of MPs in EU/EEA

is 145,000 tons). These regulations were discussed with the Member States for the first time in the Registration, Evaluation, Authorization, and Restriction of Chemicals (REACH) Committee in September 2022 (https://echa.europa.eu/hot-topics/microplastics, accessed on 5 December 2020).

The following are some of the proposed regulatory measures for controlling the release of intentionally added MNPs: (i) prohibit the products containing MNPs on placing in the markets; (ii) restrict the use of MNPs in natural/biodegradable polymers; (iii) use products without foreseeable microplastics release; (iv) recommend product labelling to minimize the release of MNPs; and (v) implement mandatory reporting requirements on the identification, description of use, tonnage, and the release of MNPs [213]. The classification of NPs is controversial, and there are three possibilities to regulate NPs by considering them as MPs, nanomaterials, or polymers [214]. Based on the size (i.e., <100 nm), NPs can be considered as "nanomaterials." However, NPs mainly contain polymeric substances and could be regulated under "polymers." However, polymers are under the exemption to register under REACH. More importantly, we do not have sufficient analytical techniques for detecting materials of <100 nm size [215].

Globally, specific regulations are under preparation. The United National Environmental Assembly (UNEA) has hosted a meeting with 175 countries for a legal binding to combat plastic pollution by 2024, and the resolution is called the "International Plastic Treaty." The draft is scheduled to be completed by the end of 2024 [216]. According to the Microbead-Free Waters Act (MFWA) of 2015, the USA passed a regulation that prohibits the manufacturing, packaging, and distributing of cosmetics containing plastic microbeads because plastic microbeads are a subset of microplastics [217]. This new law applies to both cosmetics and non-prescription drugs (e.g., toothpaste). In Canada, plastic-manufactured items have been added to the list of toxic substances under the regulation of the "Canadian Environment Protection Act's Schedule 1" [218]. According to this act, the Canadian government released draft regulations in December 2021 to ban single-use plastics (i.e., grocery/plastic bags, stir sticks, beverage six-pack holders, cutlery/plates, straws, and food packaging materials). Senate Bill 1422 (SB1422) of California focuses on establishing standards for analytical methods, lab accreditations, and provisions for health-based guidance levels [219]. This initiation monitors MPs in drinking water for four years.

The limited regulations for controlling MPs in the environment are aimed at plastic products used extensively in sectors other than agriculture. In addition, there is much focus on aquatic environments, and it is hard to find regulations to control MPs in soils. Currently, no restrictions exist for the MNPs in the agricultural sectors. This situation implies that additional investigations are greatly warranted for more understanding about MNPs and their behavior in the soil environment, especially in the agricultural ecosystem.

## 9. Conclusions and Prospects

The intensification of industrialization led to a higher use of intentional, and an accumulation of unintentional, plastics in the agroecosystems. There is a high level of burden of MNPs in agricultural soils, with their quantities detected up to 43,000 particles $kg^{-1}$ [13], and the rate of burden is expected to be a maximum of 300,000 tons of MNPs $yr^{-1}$ [54]. Both agricultural practices [44,45] and amendments [12,44,46,47] are the principal causes of soil contamination by MNPs. The available technologies are not 100% effective in removing MNPs from agricultural inputs, especially from biosolids [54]. The efficient WWT technologies can remove ~99% of MNPs from wastewater [44] and leave them in the sludge, which takes the MNPs from the aquatic system to the soil systems through agricultural practices. MNPs are unaffected for a reasonably long period in the soil environment (e.g., PVC is unchanged after 10–35 years [74]). However, agricultural activities (e.g., mechanical tillage and crop rotation) can accelerate the fragmentation of MNPs [52], which results in the unintentional release of MNPs in the soil ecosystem. MNPs further can contaminate soils in two ways: firstly, by releasing toxic additives that they contain [108], and secondly, by acting as vectors for several contaminants [77,118]. Likewise, MNPs interact with

soil biota in a complex environment that contains several contaminants, including but not limited to dioxins, PAHs, HMs, phthalates, BFRs, BPA, pharmaceuticals, agrochemicals, and engineered nanomaterials (Figure 5). In addition, aging causes several changes in MNPs in the soil ecosystem [125]. Therefore, the results of investigations carried out with the pristine forms of MNPs [158] are not comparable with the scenarios that occur in agroecosystems by MNPs. According to laboratory analyses, MNPs have exhibited positive [146,171,172] and negative [30,168,169,176,197,200,201] effects, in addition to having no effects [199], against the physical/chemical properties of soil, soil microflora, and invertebrates. The mixed impact of MNPs against abiotic and biotic factors is attributed to a wide range of variations in the experimental conditions in different investigations, such as MNP type and dose, soil types, and incubation time. MNPs exhibited adverse effects on plants either directly by direct phytotoxicity [189,197,200–204] or indirectly by altering the soil physical/chemical characteristics [177] and soil biota [31,158].

No stringent guidelines are currently available for the control of MNPs in the agricultural soils, which can be attributed to the ambiguity in the classification of MPs and NPs, unawareness about the intentional and unintentional release of MNPs from the source polymers, and uncertain health impacts. Since intensive modern agriculture relies heavily on many plastic products, there is a stronger need to gather data on the extent of contamination and the health hazards through the movement of MNPs into plants and into human and animal food. As of now, the following are the identified knowledge gaps:

- There is an urgent need to develop and standardize methods for collecting, extracting, identifying, and quantifying MNPs in agricultural soils. Due to the lack of standardized procedures, the estimation of MNPs, in terms of abundance and diversity, in different soils and regions shows significant variations and are difficult to compare or interpret. The standardized methods will also help to estimate residence times and transformations.

- Long-term studies on the effects of MNPs are needed in different soil types containing other contaminants instead of studying with pristine MNPs. The results of such investigations will help to develop prediction models and scenarios on the accumulation rates, toxicological effects, and impacts on soil health.

- There are many unresolved issues concerning the definition, sampling, ecotoxicities, and entry into the food chain of MNPs. Even though some details are available on MPs, information related to NPs is very limited. There is a strong need to determine the effects of NPs on food safety and nutritional quality.

- Future investigations should focus on human health risk assessment due to the exposure to MPs and NPs in soil via trophic transfers. Insights from these studies will help the global environmental and health agencies to set regulatory guidelines for the mitigation of plastic pollution, and to develop nature-based solutions as plastic alternatives in the agroecosystems.

- Shifting to the circular economy model for the production and use of plastics needs to be achieved across all stages of the agricultural value chain. Plastic wastes should be a source of raw materials so that the use of virgin plastics can be substantially reduced. Single-use plastics with the linear model of the "take–make–dispose" approach have detrimental effects on all natural resources.

- There is a need to develop the code of conduct and international conventions considering the life cycle of a plastic product, from its design to waste management at the end of life. Most agricultural plastic products are damaged, degraded, or discarded as "leaked plastics" in the environment [220]. The accumulation of leaked plastics will increase in agroecosystems unless measures are introduced to recycle and remove the damaged and degraded plastic products. The flows and fates of agricultural plastics should be based on the code of conduct developed involving producers, users, regional bodies, and government regulatory bodies.

- There are national, regional, and international policy and legal measures (about 291 between 2000 and 2019) for the manufacturing of plastic products, and for the single-

use plastics, but there are no "globally binding targets or instruments" to reduce plastic pollution [221]. Since scientific investigations suggest plausible risks, the precautionary principle, in addition to the "polluter pays" principle, must be applied for plastics in agroecosystems. Good agricultural practices for reducing plastic pollution require innovative technical solutions as alternatives to plastics, and regulatory measures, along with behavioral changes by the farmers, for improved sustainability of the agroecosystems.

- Ultra-performance Liquid chromatography-based Quadrupole Time-of-Flight Mass Spectroscopy (UPLC-QTOF-MS) [222] and Aptamer-based biosensors [223] are the promising technologies for environmental monitoring of the emerging pollutants.
- Use of alternative products, biodegradable plastics, and recycling and disposal of plastic wastes significantly mitigate the farmland soil burden by MNPs. Alternative plastic mulch-like materials include biodegradable plastic mulch, cellulose-based paper mulch, organic mulches such as straw-based mulches, strip tilling, deep compost mulch, woodchips, wool mulch (i.e., woolch), etc. [224]. According to a case study, alternatives to plastic mulches were found to be effective in suppressing weed cover and yielding taller plants over the un-mulched controlled plots [225].

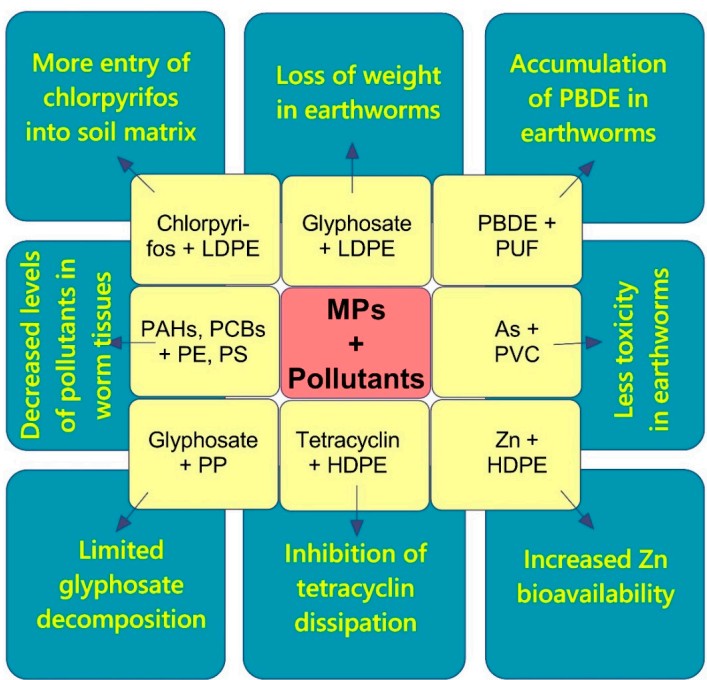

**Figure 5.** Impact of MPs in the presence of other environmental pollutants in soil [226]. LDPE—Low-density polyethylene, PBDE—Polybrominated diphenyl ethers, PUF—Polyurethane foam, PAHs—Polycyclic aromatic hydrocarbons, PCBs—Polychlorinated biphenyls, PE—Polyethene, PS—Polystyrene, PVC—Polyvinyl chloride, PP—Polypropylene, HDPE—High-density polyethylene.

**Author Contributions:** Conceptualization, N.R.M., B.R., K.V. and M.M.; Methodology, N.R.M.; Writing—Original draft preparation, N.R.M. and B.R.; Writing—review and editing, K.V., B.R., N.R.M., T.K. and M.M. All authors have read and agreed to the published version of the manuscript.

**Funding:** This research received no external funding.

**Institutional Review Board Statement:** Not applicable.

**Informed Consent Statement:** Not applicable.

**Data Availability Statement:** Not applicable.

**Conflicts of Interest:** The authors declare no conflict of interest.

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
