# Peer review of "Do Microplastics and Nanoplastics Pose Risks to Biota in Agricultural Ecosystems?"

_soilsystems, doi:10.3390/soilsystems7010019_

Round 1

Reviewer 1 Report

Review Report on the Review Article: Do Micro- and Nano plastics Pose Risks to Biota in Agricultural Ecosystems?

1.     General

The purpose of this review article was to provide comprehensive insights on the sources of micro- and nano plastics in agricultural soils, its nature, transformation, release, fate and its influences on soil physicochemical properties as well as toxicity to soil biota (microbial diversity, invertebrates and plants) including regulations for mitigations of MNPs in agroecosystems, pointing out knowledge gaps and future research directions.

 ·       The review article provided critically reviewed and up-to-date information (majority with recent publications) in a very comprehensive manner on plastic soil pollution (point and non-point sources) and its impact on soil physicochemical and biological properties, which is very clear and of relevance to the field.  

 ·       The approach, content and analyses and organization of the review is scientifically correct and to the standard.

 ·       Knowledge gap is clearly identified with well-articulated research areas for further considerations.

 ·       This review article is very timely, relevant and it is of interest to the scientific community in the field.

 ·       The descriptions and conclusions provided are coherent and supported by relevant citations.

 ·       The figures and tables provided are all appropriate and well-illustrated.

 ·       Almost all cited references are recent publications (more than 58% are publications with the last five years).

Specific comments

·       Section 3: Line 293-309- while the review on the transformation of the various MNPs (PE, PP, PVC, PET, PS, etc.) is done well, current knowledge on advances in biodegradation processes/technologies under aerobic and anaerobic, in situ and ex situ conditions as well as degradability rates are not explicitly provided.

·       Under conclusion and prospects: Line 757: …..available technologies are not 100% effective in removing MNPs from agricultural inputs, specially from biosolids…..What about prospects for technical advances to manage these emerging pollutants in the future?

·       Knowledge gaps: line 807-809: ….shifting to circular economy model for the production and use of plastics………. This is a bit vague considering that MNPs are nearly non-biodegradable. This might only be applicable to source reduction and or shift to biodegradable plastics! Some clarity is also needed here.

Author Response

Reviewer # 1

General

The purpose of this review article was to provide comprehensive insights on the sources of micro- and nano plastics in agricultural soils, its nature, transformation, release, fate and its influences on soil physicochemical properties as well as toxicity to soil biota (microbial diversity, invertebrates and plants) including regulations for mitigations of MNPs in agroecosystems, pointing out knowledge gaps and future research directions.

The review article provided critically reviewed and up-to-date information (majority with recent publications) in a very comprehensive manner on plastic soil pollution (point and non-point sources) and its impact on soil physicochemical and biological properties, which is very clear and of relevance to the field.  

The approach, content and analyses and organization of the review is scientifically correct and to the standard.

Knowledge gap is clearly identified with well-articulated research areas for further considerations.

This review article is very timely, relevant and it is of interest to the scientific community in the field.

The descriptions and conclusions provided are coherent and supported by relevant citations.

The figures and tables provided are all appropriate and well-illustrated.

Almost all cited references are recent publications (more than 58% are publications with the last five years).

Authors’ response: We thank the Referee for the positive comments, encouragement, and suggestions. As desired, we have revised the manuscript with the relevant information.

Specific comments

Comment: Section 3: Line 293-309- while the review on the transformation of the various MNPs (PE, PP, PVC, PET, PS, etc.) is done well, current knowledge on advances in biodegradation processes/technologies under aerobic and anaerobic, in situ and ex situ conditions as well as degradability rates are not explicitly provided.

Authors’ response: As required, following are the details included in the revised version.

The removal constants (K) for PP in Rhodococcus sp. str. 36 and Bacillus sp. str. 27 were 0.002 and 0.001 day–1, respectively [83].” (Lines: 301-302)

The dry weight of PE MPs was reduced by 14.70% after 60 days in non-carbonaceous basal medium inoculated with Paenibacillus sp. [84].” (Lines: 304-306)

The mechanical properties of low-density polyethylene (LDPE) were changed by 27% after 150 days of treatment by Pleurotus ostreatus [86].” (Lines: 311-312)

Comment: Under conclusion and prospects: Line 757: …..available technologies are not 100% effective in removing MNPs from agricultural inputs, specially from biosolids…..What about prospects for technical advances to manage these emerging pollutants in the future?

Authors’ response: Additional details were provided in Section 8 (i.e., Regulatory Guidelines for Mitigation of MNPs) as shown below:

Ultra-performance Liquid chromatography-based Quadrupole Time-of-Flight Mass Spectroscopy (UPLC-QTOF-MS) [222] and Aptamer-based biosensors [223] are the promising technologies for environmental monitoring of emerging pollutants.

Use of alternative products, biodegradable plastics, and recycling and disposal of plastic wastes significantly mitigate the farmland soil burden by MNPs. Alternative plastic mulch-like materials include biodegradable plastic mulch, cellulose-based paper mulch, organic mulches such as straw-based mulches, strip-tilling, deep compost mulch, woodchips, wool mulch (i.e., woolch), etc. [224]. According to a case study, alternatives to plastic mulches were found to be effective in suppressing weed cover and yielding taller plants over the un-mulched controlled plots [225].” (Lines: 845-854)

Comment: Knowledge gaps: line 807-809: ….shifting to circular economy model for the production and use of plastics………. This is a bit vague considering that MNPs are nearly non-biodegradable. This might only be applicable to source reduction and or shift to biodegradable plastics! Some clarity is also needed here.

Authors’ response: Following is the corrected sentence.

“Plastic wastes should be a source of raw materials, so that the use of virgin plastics can be substantially reduced.” (Lines: 824-826)

Reviewer 2 Report

Overall comment

This is a well-writen manuscript. It covers the intended areas adquetly while citing recent publications. Therefore, I recommend it is suitable for publish in Soil Systems. 

Specific comments 

1. Delete keywords that are already appearing in the title. 

2.  "Eco-toxin" term should be replaced with "Eco-toxicant" ? Toxin refers to biological origin.

3. Provide examples for the "Other 17%" plastic use category.

4. Show "Interestingly, China is the world's largest producer of plastics (31% of the overall production), and other Asian countries, including India, synthesize about half the plastics in the world" after the global plastic production sentence (i.e., Line 68-69).

5. Provide meaning of abbreviations (e.g., CAGR) appropriately, following the journal guidelines. 

6. Authors have classified plastics into Micro and Nano in Line 85. How about macroplastics?

7. Authors have provided definition for microplastics in two places. Delete one. 

8. Use abbreviations and units in a consistent way, ton vs t. 

9. "...challenging to get" or "challenging to estimate"

10. Better to integrate short meaning for the "Damköhler number (DN)"

11. A description/ paragraph should be appeared before the FIgure 5.

12. Better to include a description covering mitigation strategies in soil-MNPs pollution 

Author Response

Reviewer # 2

Overall comment

This is a well-written manuscript. It covers the intended areas adequately while citing recent publications. Therefore, I recommend it is suitable for publish in Soil Systems. 

Authors’ response: We are highly grateful to the Referee for the constructive feedback.

Specific comments 

Comment: Delete keywords that are already appearing in the title. 

Authors’ response: Repeated keyword was deleted.

Comment: "Eco-toxin" term should be replaced with "Eco-toxicant" ? Toxin refers to biological origin.

Authors’ response: Corrected as “Ecotoxicity”. (Line: 58)

Comment: Provide examples for the "Other 17%" plastic use category.

Authors’ response: Details were provided as shown below:

…….; this includes plastics for furniture, medical applications, machinery and mechanical engineering, technical parts, etc.) [7].” Lines: 73-75)

Comment: Show "Interestingly, China is the world's largest producer of plastics (31% of the overall production), and other Asian countries, including India, synthesize about half the plastics in the world" after the global plastic production sentence (i.e., Line 68-69).

Authors’ response: The sentence has been shifted to the appropriate place.

“Interestingly, China is the world's largest producer of plastics (31% of the overall production), while other Asian countries, including India, synthesize about half the amounts of plastics in the world.” (Lines: 68-70)

Comment: Provide meaning of abbreviations (e.g., CAGR) appropriately, following the journal guidelines. 

Authors’ response: Abbreviation has been elaborated.

Intensive global production of resins and fibers, with an increase of 8.40% in compound annual growth rate (CAGR) from 1950 (2 Mt) to 2015 (380 Mt)) [9], resulted in the release of vast quantities of plastic wastes from different sectors.” (Lines: 79-81)

Comment: Authors have classified plastics into Micro and Nano in Line 85. How about macroplastics?

Authors’ response: The following mention has also been made of ‘macroplastics’.

Plastics of ˃5 mm size are considered as ‘macroplastics’.” (Line: 87)

Comment: Authors have provided definition for microplastics in two places. Delete one. 

Authors’ response: Deleted as suggested.

Comment: Use abbreviations and units in a consistent way, ton vs t. 

Authors’ response: Abbreviations and units were presented consistently in the revised version.

Comment: "...challenging to get" or "challenging to estimate"

Authors’ response:  Corrected.

Comment: Better to integrate short meaning for the "Damköhler number (DN)"

Authors’ response: Details were provided as shown below:

“…… Damköhler number (DN, defined as dimensionless ratio of a characteristic liquid residence time to the reaction time) [122]. (Lines: 393-394)

Comment: A description/ paragraph should be appeared before the Figure 5.

Authors’ response: Figure 5 has been shifted to the appropriate place to ensure its appearance after citing the figure in the text.

Comment: Better to include a description covering mitigation strategies in soil-MNPs pollution 

Authors’ response: Mention has been made about the difficulty in mitigation of MNPs pollution in the soil as follows.

“No stringent guidelines are currently available for the control of MNPs in the agricultural soils, which can be attributed to the ambiguity in the classification of MPs and NPs, unawareness about the intentional and unintentional release of MNPs from the source polymers, and uncertain health impacts.” (Lines: 793-796).

Reviewer 3 Report

Line 86: The definition of nanoplastics is often controversial. I suggest relying on this article: Hartmann, N. B., Huffer, T., Thompson, R. C., Hassellöv, M., Verschoor, A., Daugaard, A. E., ... & Wagner, M. (2019). Are we speaking the same language? Recommendations for a definition and categorization framework for plastic debris.

Line 97: Figure 1b is not clear. Please, better explain in the text.

Line 98: Figure 1c. What are the differences between pieces and particles? Explain in the text.

Line 100-108: the sentence is too long and difficult to connect logically. Please rephrase.

Line 162-163: There is no logical connection between the two sentences.

Line 169: It would be better to use units of measurement in the international system and thus use metres instead of yards.

Line 190: See previous comment (line 169).

Line 200: Please, clarify the meaning of dwt.

Line 205: I assume the meaning is that 63% MP and 92% NP in the soil are synthetic fibres, is this correct? Otherwise change the sentence.

Line 207: in reference [44] are reported 3.50 p g-1, meaning that are 3.50 particles / gram. I do not understand what is meant here as pg.

Line 208: Again, it is not clear what pg means. Moreover, check the accuracy of the data reported.

Line 233: Since this is the first time it appears, please explain polyethylene (PE)

Line 236: Insert (PET)

Line 300: You claim that fungi degrade MNPs more efficiently than bacteria, but in the above examples you mention only one fungus. It would be better to avoid this antithesis.

Table 1: This table represents a gross oversimplification compared to the microorganisms reported to be involved in the degradation of plastics according to recent literature.

Chapter 4 seems shallow. A table with the main additives used could be included.

Line 387: It might be useful to better specify what DN is and how it is calculated so that it is easier to understand the subsequent text.

Line 415: PAH Explicit polycyclic aromatic hydrocarbons

Figure 2 is not clear and adds nothing to the text.

Line 455: The figure 3 does not refer to changes in biological properties, nor is it the subject of this paragraph. Perhaps it is better to put the sentence in paragraph 7.

Figure 3 c: what is the concentration of MPs that causes this decrease?

Line 474: (Figure 3d) is missing.

Line 483: see comment line 169.

Figure 4c and 4d do not refer to this sub-paragraph. It would be better to insert the figure elsewhere or divide it into its parts.

7.1 Microbial Diversity: The aspect of fungal communities, which are mentioned only marginally on line 564, should also be explored in this paragraph.

7.3. Plants: It would be desirable for the authors to not only report data from the literature, but to give a contextualised and critical view of it. Crops, and thus plants, are the focal point of the agriculture that is the subject of this review. It would be interesting to amend the paragraph by highlighting the contextualisation with the title of the review.

Regulatory Guidelines for Mitigation of MNPs: Interesting chapter, but out of context with the title of the review. Agriculture.

In general, the focus on the agricultural environment is sometimes lost in the text and other areas get a little off topic.

Line 755: see line 169.

Figure 5: I would only leave figure 5a in the conclusions. I would move 5b to section 7.2 together with figure 4c. Figure 5c seems redundant for conclusions.

Author Response

Reviewer # 3

Comment - Line 86: The definition of nanoplastics is often controversial. I suggest relying on this article: Hartmann, N. B., Huffer, T., Thompson, R. C., Hassellöv, M., Verschoor, A., Daugaard, A. E., ... & Wagner, M. (2019). Are we speaking the same language? Recommendations for a definition and categorization framework for plastic debris.

Authors’ response: We thank the Reviewer for the suggestion. As per the reference suggested, there exists a clear discrepancy about the size of nanoparticles. We, therefore, wish to retain the present description of <100 nm size for nanoplastics.

Comment - Line 97: Figure 1b is not clear. Please, better explain in the text.

Authors’ response: Following additional information has been provided in the revised version.

The per cent of MNPs generated and accumulated (i.e., 75.60%) in municipal solid waste landfills in the USA (reported in 2018, Figure 1b) and the occurrence of MNPs (either in the form of pieces or particles) in farmland soils in different countries (Figure 1c) suggest that MNPs pose a great threat to agricultural ecosystems.” (Lines: 97-100)

Comment - Line 98: Figure 1c. What are the differences between pieces and particles? Explain in the text.

Authors’ response: No specific definitions are available for the terms ‘pieces’ and ‘particles’. Both the words are interchangeably used to mean the same. Hence, we used the version “pieces or particles” in the revised version.

Comment - Line 100-108: the sentence is too long and difficult to connect logically. Please rephrase.

Authors’ response: The sentence has been rephrased as suggested.

Comment - Line 162-163: There is no logical connection between the two sentences.

Authors’ response: Use of plastic film mulch is a great revolution in agriculture, which is considered as a ‘white revolution’. However, the use of large amounts of residual plastic film has detrimental effects on soil structure, water and nutrient transport and crop growth, thereby disrupting the agricultural environment and reducing crop production. Thus, there exists a clear link between ‘white revolution’ and ‘white pollution’.

Comment - Line 169: It would be better to use units of measurement in the international system and thus use metres instead of yards.

Authors’ response: It is not yards, it is the year (yr‒1).

Comment - Line 190: See previous comment (line 169).

Authors’ response: It is the year.

Comment - Line 200: Please, clarify the meaning of dwt.

Authors’ response: The expansion, “dry weight” has now been provided for the abbreviation ‘dwt’. (Line: 202)

Comment - Line 205: I assume the meaning is that 63% MP and 92% NP in the soil are synthetic fibres, is this correct? Otherwise change the sentence.

Authors’ response: These values for MP and NP taken from two different studies are correct.

Comment - Line 207: in reference [44] are reported 3.50 p g-1, meaning that are 3.50 particles / gram. I do not understand what is meant here as pg.

Authors’ response: It is the ‘picogram’. The word ‘picogram’ has been included now.

Comment - Line 208: Again, it is not clear what pg means. Moreover, check the accuracy of the data reported.

Authors’ response: It is the picogram.

Comment - Line 233: Since this is the first time it appears, please explain polyethylene (PE).

Authors’ response: Corrected as per the suggestion.

Comment - Line 236: Insert (PET)

Authors’ response: Corrected accordingly.

Comment - Line 300: You claim that fungi degrade MNPs more efficiently than bacteria, but in the above examples you mention only one fungus. It would be better to avoid this antithesis.

Authors’ response: We agree with this suggestion, and the corrected sentence is as follows.

Fungal species are efficient in the degradation of MNPs; …” (Line: 306)

Comment - Table 1: This table represents a gross oversimplification compared to the microorganisms reported to be involved in the degradation of plastics according to recent literature.

Authors’ response: The main purpose of presenting Table 1 is to highlight the efficiency of microbial species in degrading different MNPs in soil. Therefore, only four parameters were included, i.e., type of MNP, microbial species, incubation time and weight loss. As the experimental conditions were diverse among the studies, it is therefore difficult to bring uniformity for different parameters in a single table.

Comment - Chapter 4 seems shallow. A table with the main additives used could be included.

Authors’ response: The available information on release and fate of additives from MNPs is fragmentary. Hence, instead of including a table, we provided the necessary details in the text (>600 words) by including the available 11 relevant references.

Comment - Line 387: It might be useful to better specify what DN is and how it is calculated so that it is easier to understand the subsequent text.

Authors’ response: The abbreviation, DN, has been elaborated now.

“……Damköhler number (DN), which is the dimensionless ratio of a characteristic liquid residence time to the reaction time) [122]. (Lines: 393-394)

Comment - Line 415: PAH Explicit polycyclic aromatic hydrocarbons

Authors’ response: Corrected accordingly (Lines: 421-422).

Comment - Figure 2 is not clear and adds nothing to the text.

Authors’ response: The following sentence has been included now.

To understand the distribution and recovery of pesticides in the presence of MPs, a soil column experiment was conducted by Ramos et al. [136]. They found that the adsorption of trifluralin, procymidone and chlorpyrifos to PE were 98.90, 95.90, and 98.60%, respectively (Figure 2), implying the propensity of PE to act as vectors to carry different organic contaminants in the soil ecosystem.” (Lines: 423-427)

Comment - Line 455: The figure 3 does not refer to changes in biological properties, nor is it the subject of this paragraph. Perhaps it is better to put the sentence in paragraph 7.

Authors’ response: We thank the Reviewer for the valuable observation. 

Figure 3 has now been cited at the appropriate place. (Line: 453)

Comment - Figure 3 c: what is the concentration of MPs that causes this decrease?

Authors’ response: The concentrations of MPs that caused the decrease have been included in the figure caption as shown below.

“(C) Response of soil properties to MPs at concentrations ranging from 0.01 to 640,000 mg kg–1 (meta-analysis data from Gao et al. [20]). (Lines: 500-501)

Comment - Line 474: (Figure 3d) is missing.

Authors’ response: Figure 3d cited in Lines: 477-478 now.

Comment - Line 483: see comment line 169.

Authors’ response: It is the year.

Comment - Figure 4c and 4d do not refer to this sub-paragraph. It would be better to insert the figure elsewhere or divide it into its parts.

Authors’ response: The Figure 4 has been inserted after citing Figure 4d in the Section 7.

Comment - 7.1 Microbial Diversity: The aspect of fungal communities, which are mentioned only marginally on line 564, should also be explored in this paragraph.

Authors’ response: Following additional information has been provided in the revised version.

When metal ion blended PP were added to the cultures of Phanerochaete chrysosporium NCIM 1170 and Engyodontium album MTP091 for one year, the gravimetric weight loss and thermos gravimetric analysis (TGA) weight loss were in the range of 9.42-18.80 and 57-79%, respectively [98]. Alternaria sp., Penicilium section Lanata-Divaricata and Aspergillus section flavi were found to utilize two polyester polyurethane as sole carbon source [106]. Aboveground biomass and colonization of AM (arbuscular mycorrhizal) fungi were increased under polyester microfiber addition [171].” (Lines: 567-573)

Comment - 7.3. Plants: It would be desirable for the authors to not only report data from the literature, but to give a contextualised and critical view of it. Crops, and thus plants, are the focal point of the agriculture that is the subject of this review. It would be interesting to amend the paragraph by highlighting the contextualisation with the title of the review.

Authors’ response: In fact, in the first paragraph of Section 7.3, there are details about the uptake mechanisms of MNPs by different food crops, and different analytical techniques that were used for the analysis of MNPs in plant materials. In the 2nd and 3rd paragraphs, in-depth analysis was done on the phytotoxicity of MNPs. Therefore, the main aim in presenting this section was to provide the details on the entry, root uptake and translocation of MNPs as well as the adverse effects of MNPs on plant functionalities.

A summary of the Section 7.3 has been provided in the revised version, as detailed below:

“The above insights clearly imply that MNPs in farmland soils are subjected to root uptake and translocation into food crops, posing a significant threat to the terrestrial food chain by MNPs in farmlands. However, technical barriers in the quantification of MNPs limit the mitigation of MNPs' entry into farmland soils and their further dissipation in the agroecosystems.” (Lines: 707-712)

Comment - Regulatory Guidelines for Mitigation of MNPs: Interesting chapter, but out of context with the title of the review. Agriculture.

In general, the focus on the agricultural environment is sometimes lost in the text and other areas get a little off topic.

Authors’ response: To reflect the title of the present review, following additional information has been provided in the revised version.

“Plastic wastes should be a source of raw materials, so that the use of virgin plastics can be substantially reduced.” (Lines: 824-826)

“Ultra-performance Liquid chromatography-based Quadrupole Time-of-Flight Mass Spectroscopy (UPLC-QTOF-MS) [222] and Aptamer-based biosensors [223] are the promising technologies for environmental monitoring of the emerging pollutants.

Use of alternative products, biodegradable plastics, and recycling and disposal of plastic wastes significantly mitigate the farmland soil burden by MNPs. Alternative plastic mulch-like materials include biodegradable plastic mulch, cellulose-based paper mulch, organic mulches such as straw-based mulches, strip tilling, deep compost mulch, woodchips, wool mulch (i.e., woolch), etc. [224]. According to a case study, alternatives to plastic mulches were found to be effective in suppressing weed cover and yielding taller plants over the un-mulched controlled plots [225].” (Lines: 845-854)

Comment - Line 755: see line 169.

Authors’ response: It is the year.

Comment - Figure 5: I would only leave figure 5a in the conclusions. I would move 5b to section 7.2 together with figure 4c. Figure 5c seems redundant for conclusions.

Authors’ response: We thank the Reviewer for the suggestion. Accordingly, we deleted Figures 5b and 5c.

Reviewer 4 Report

The authors have very carefully addressed the issue of plastics in soil, in all the different aspects (chemical-physical properties, morphology, adsorption of contaminants, degradation, toxicity, etc.) necessary to assess the environmental risk related to their presence in soils.

The review provides a very detailed picture of the issue and can give good guidance to the scientific community. Some minor comments/requests for revisions are given below. 

Pag. 2 Line 56: suggest eliminating “Like pesticide” at the end of the sentence.

Lines 59-60:

“Thus, microplastics (MPs) represent a diverse suite of contaminants [4]”

Please, explain why the authors make a similar sentence. The different morphology o chemical composition of plastics makes plastics contaminants. The situation from the chemical point of view is quite complicated because of the presence of additives, or the sorption of other types of contaminants dispersed in the systems, and even by the degradation of plastics, which could introduce soluble and insoluble fractions of polymers.

So, the sentence must be deepened, because is scarcely clear or eliminated because the concept is very well explained in paragraphs 2 and 3.

Lines 63-66 The sentence might be removed. I would avoid historical information about materials.

'Parkesine' was the first man-made plastic, invented by Alexander Parkes in 1862, 63 while celluloid was by John Wesley Hyatt in 1869. In 1907, 'Bakelite,' the first synthetic 64 plastic, was created by Leo Baekeland, who coined the term plastic,' which originally 65 meant 'pliable and easily shaped' and is now a chain of synthetic polymers [5].

Concerning bibliographical references, a lengthy preparatory study is evident in the work of writing the manuscript, but, since the field of plastics is very vast and constantly evolving, it would be advisable to carefully select the bibliographical notes limiting the period of publication in the last 10 years, to make the use of concepts and reading of the review more effective.

Author Response

Reviewer # 4

Comment: The authors have very carefully addressed the issue of plastics in soil, in all the different aspects (chemical-physical properties, morphology, adsorption of contaminants, degradation, toxicity, etc.) necessary to assess the environmental risk related to their presence in soils.

The review provides a very detailed picture of the issue and can give good guidance to the scientific community. Some minor comments/requests for revisions are given below.

Authors’ response: We very much appreciate the positive feedback from the Reviewer.

Comment - Pag. 2 Line 56: suggest eliminating “Like pesticide” at the end of the sentence.

Authors’ response: This sentence was deleted in the revised version as the definition for MNPs was provided at two places.

Comment - Lines 59-60: “Thus, microplastics (MPs) represent a diverse suite of contaminants [4]”

Please, explain why the authors make a similar sentence. The different morphology o chemical composition of plastics makes plastics contaminants. The situation from the chemical point of view is quite complicated because of the presence of additives, or the sorption of other types of contaminants dispersed in the systems, and even by the degradation of plastics, which could introduce soluble and insoluble fractions of polymers.

So, the sentence must be deepened, because is scarcely clear or eliminated because the concept is very well explained in paragraphs 2 and 3.

Authors’ response: For a better clarity, the two sentences have been restructured as follows.  

“Microplastics (MPs) of different morphology, colour, and ecotoxicity that come from various organic polymers and blended with different additives represent a diverse suite of contaminants [4].” (Lines: 58-60)

Comment - Lines 63-66: The sentence might be removed. I would avoid historical information about materials.

'Parkesine' was the first man-made plastic, invented by Alexander Parkes in 1862, 63 while celluloid was by John Wesley Hyatt in 1869. In 1907, 'Bakelite,' the first synthetic 64 plastic, was created by Leo Baekeland, who coined the term plastic,' which originally 65 meant 'pliable and easily shaped' and is now a chain of synthetic polymers [5].

Authors’ response: Based on the comment, the above sentence has been simplified as shown below.

The term 'plastic', which originally meant 'pliable and easily shaped', is now used for a chain of synthetic polymers [5], but ‘plastic’ is historically known as ‘celluloid’ and ‘bakelite’. (Lines: 63-65)

Comment: Concerning bibliographical references, a lengthy preparatory study is evident in the work of writing the manuscript, but, since the field of plastics is very vast and constantly evolving, it would be advisable to carefully select the bibliographical notes limiting the period of publication in the last 10 years, to make the use of concepts and reading of the review more effective.

Authors’ response: Mention may be made that around 60% of the references cited belonged to the last five years. It is evident from the Figure 1a that there is a tremendous increase in the publications related to ‘MPs versus soil’ during the period of 2018‒2022.

Reviewer 5 Report

Dear Authors,

Your scientific work is very interesting because it presents problems related to the presence of nano- and microplastics in agrosystems. In my opinion, the manuscript is carefully prepared. The layout of the articles became standard for research papers. However, I have a few comments and remarks:

L 16: Please explain the abbreviation Mt.

The font throughout the text should be in black.

Figures 3 and 5 are illegible. Please enlarge the figures.

In my opinion, the manuscript is well written. The above comments do not reduce the value of the article.

The language appears to be correct, but I don't feel qualified to judge about the English language and style.

Good luck!

Sincerely yours

Reviewer

Author Response

Reviewer # 5

Comment: Your scientific work is very interesting because it presents problems related to the presence of nano- and microplastics in agrosystems. In my opinion, the manuscript is carefully prepared. The layout of the articles became standard for research papers. However, I have a few comments and remarks:

Authors’ response: We thank the Reviewer for the positive feedback and encouragement.

Comment - L 16: Please explain the abbreviation Mt.

Authors’ response: The abbreviation ‘Mt’ has been elaborated as ‘Megatons’. (Line: 18)

Comment: The font throughout the text should be in black.

Authors’ response: Throughout the manuscript, other colours have been removed now.

Comment: Figures 3 and 5 are illegible. Please enlarge the figures.

Authors’ response: As per the suggestion, Figure 3 and 5 have been enlarged.

Comment: In my opinion, the manuscript is well written. The above comments do not reduce the value of the article. The language appears to be correct, but I don't feel qualified to judge about the English language and style.

Authors’ response: Once again, we take this opportunity to thank the Reviewer for the appreciation.

Round 2

Reviewer 3 Report

No other comment. Authors have optimally edited the text

Author Response

We very much appreciate Reviewer 3 for approving our earlier revised version. As per the suggestion of Reviewer 3, we performed the required check for grammar and spelling.